# Towards Robust Fidelity for Evaluating Explainability of Graph Neural Networks

**Xu Zheng**[1]*, **Farhad Shirani**[1]*, **Tianchun Wang**[2], **Wei Cheng**[3], **Zhuomin Chen**[1],
**Haifeng Chen**[3], **Hua Wei**[4], **Dongsheng Luo**[1]✉

[1]School of Computing and Information Sciences, Florida International University, US
[2]College Information Sciences and Technology, The Pennsylvania State University, US
[3]NEC Laboratories America, US
[4]School of Computing and Augmented Intelligence, Arizona State University, US
`{xzhen019,fshirani,zchen051,dluo}@fiu.edu`
`tkw5356@psu.edu`
`{weicheng,haifeng}@nec-labs.com`
`hua.wei@asu.edu`

## Abstract

Graph Neural Networks (GNNs) are neural models that leverage the dependency structure in graphical data via message passing among the graph nodes. GNNs have emerged as pivotal architectures in analyzing graph-structured data, and their expansive application in sensitive domains requires a comprehensive understanding of their decision-making processes — necessitating a framework for GNN explainability. An explanation function for GNNs takes a pre-trained GNN along with a graph as input, to produce a 'sufficient statistic' subgraph with respect to the graph label. A main challenge in studying GNN explainability is to provide fidelity measures that evaluate the performance of these explanation functions. This paper studies this foundational challenge, spotlighting the inherent limitations of prevailing fidelity metrics, including $Fid_+$, $Fid_-$, and $Fid_\Delta$. Specifically, a formal, information-theoretic definition of explainability is introduced and it is shown that existing metrics often fail to align with this definition across various statistical scenarios. The reason is due to potential distribution shifts when subgraphs are removed in computing these fidelity measures. Subsequently, a robust class of fidelity measures are introduced, and it is shown analytically that they are resilient to distribution shift issues and are applicable in a wide range of scenarios. Extensive empirical analysis on both synthetic and real datasets are provided to illustrate that the proposed metrics are more coherent with gold standard metrics. The source code is available at https://trustai4s-lab.github.io/fidelity.

## 1 Introduction

Graph Neural Networks (GNNs) have become a cornerstone for processing graph-structured data, achieving impressive outcomes across various domains such as node classification and link prediction (Kipf & Welling, 2017; Hamilton et al., 2017; Veličković et al., 2018; Scarselli et al., 2008). However, with their proliferation in sensitive sectors like healthcare and fraud detection, the demand for understanding their decision-making processes has grown significantly (Zhang et al., 2022a; Wu et al., 2022; Li et al., 2022). To address this challenge, explanation techniques have been proposed for GNNs, most commonly focusing on identifying a subgraph that dominates the model's prediction in a post-hoc sense (Ying et al., 2019; Luo et al., 2020; Yuan et al., 2021).

In the design and study of explainable GNNs, both model design and choice of evaluation metrics are important. While most efforts have primarily been made to develop new network architectures and optimization objectives to achieve more accurate explanations (Ying et al., 2019; Luo et al., 2020;

---

*These authors contributed equally to this work.
✉Corresponding author

Yuan et al., 2021), in this paper, we underscore the critical importance of choosing the right evaluation metrics for the achieved explanations. In an ideal scenario, quantitative evaluation of an explanation subgraph can be achieved by comparing it with a gold standard or ground truth explanation (Ying et al., 2019). Yet, in real-world applications, such ground truth explanation subgraphs are a rarity, often making direct comparisons impracticable. In lieu of this, *surrogate* fidelity metrics, namely $Fid_+$, $Fid_-$, and $Fid_\Delta$, have been included to gauge the faithfulness of explanation subgraphs. At its core, the intuition driving such fidelity metrics is straightforward: if a subgraph is discriminative to the model, the prediction should change significantly when it is removed from the input. Otherwise, the prediction should be maintained. Hence, $Fid_+$ is defined as the difference in accuracy (or predicted probability) between the original prediction and the new predictions of non-explanation subgraph which is obtained by masking out the explanation subgraph (Pope et al., 2019; Bajaj et al., 2021), and $Fid_-$ measures the difference between predictions of the original graph and explanation subgraph (Yuan et al., 2022). As prevailing standards, these Fidelity metrics and their variants have been widely used in existing popular platforms, such as GraphFramEx Amara et al. (2022), GraphXAI (Agarwal et al., 2023), GNNX-BENCH (Kosan et al., 2023), and DIG (Liu et al., 2021).

Although intuitively correct, we argue that the aforementioned Fidelity metrics come with significant drawbacks due to the impractical assumption that the to-be-explained model can make accurate predictions of the explanation subgraph (in $Fid_-$) or non-explanation subgraph (in $Fid_+$). This does not hold in a wide range of real-world scenarios, because when edges are removed, the resultant subgraphs might be Out Of Distribution (OOD) (Fang et al., 2023c;b). For example, in MUTAG dataset (Debnath et al., 1991), each graph is a molecule with nodes representing atoms and edges describing the chemical bonds. The functional group $NO_2$ is considered the dominating subgraph that causes a molecule to be mutagenic. The explanation subgraph only consists of 2 edges, which is much smaller than whole molecular graphs. Such disparities in properties introduce distribution shifts, putting the Fidelity metrics on shaky grounds, because of the violation of a key assumption in machine learning: the training and test data come from the same distribution (Hooker et al., 2019).

To build an evaluation foundation for eXplainable AI (XAI) in the graph domain, In this paper, we investigate robust fidelity measurements for evaluating the correctness of explanations. There are several non-trivial challenges associated with this problem. First, the to-be-explained GNN model is usually evaluated as a black-box model, which cannot be re-trained to ensure the generalization capacity (Hooker et al., 2019). Second, the evaluation method is required to be stable and ideally deterministic. As a result, complex parametric methods, such as adversarial perturbations (Hase et al., 2021; Hsieh et al., 2021), are not suitable as the results are affected by randomly initiated parameters.

We provide an information theoretic framework for GNN explainability. We define an explanation graph as a subgraph of the input that satisfies two conditions: 1) the explanation graph is an (almost) sufficient statistic of the input graph with respect to the output label, and 2) the size of the explanation graph is *small* compared to the input graph. Based on this, we provide two notions of explainability, namely, explainability of a classification task and explainability of a classifier for the task (Definitions 2 and 3). We quantify the relation between these two notions for low-error classifiers and classification tasks (Theorem 1). Computing the resulting information theoretic fidelity measure requires knowledge of the underlying graph statistics, which is not possible in most real-world scenarios of interest. We propose a generalized class of surrogate fidelity measures that are robust to distribution shift issues in a wide range of scenarios (Proposition A.4). Our contributions are summarized as follows.

- We pioneer in spotlighting the inherent shortcomings of widely accepted evaluation methodologies in the explainable graph learning domain.
- Grounded in solid theoretical underpinnings, we introduce novel evaluation metrics that are resilient to distribution shifts, enhancing their applicability in real-world contexts. This metric notably approximates evaluations conducted with ground truth explanations more closely.
- Through rigorous empirical analyses on a diverse mix of synthetic and real datasets, we validate that our approach resonates well with gold standard benchmarks.

## 2 PRELIMINARIES

**Random Graphs:** We parameterize a (random) labeled graph $G$ by a tuple $(\mathcal{V}, \mathcal{E}; Y, \boldsymbol{X}, \boldsymbol{A})$, where i) $\mathcal{V} = \{v_1, v_2, ..., v_n\}$ is the vertex set[1], ii) $\mathcal{E} \subseteq \mathcal{V} \times \mathcal{V}$ is the edge set, iii) $Y$ is the graph class label

---

[1]We use node and vertex interchangeably.

taking values from finite set of classes $\mathcal{Y}$, iv) $\boldsymbol{X} \in \mathbb{R}^{n \times d}$ is the feature matrix, where the $i$-th row of $\boldsymbol{X}$, denoted by $\boldsymbol{X}_i \in \mathbb{R}^{1 \times d}$, is the $d$-dimensional feature vector associated with node $v_i, i \in [n]$, and v) $\boldsymbol{A} \in \{0, 1\}^{n \times n}$ is the adjacency matrix. The graph parameters $(Y, \boldsymbol{A}, \boldsymbol{X})$ are generated according to the joint probability measure $P_{Y, \boldsymbol{A}, \boldsymbol{X}}$. Note that the adjacency matrix determines the edge set $\mathcal{E}$, where $A_{ij} = 1$ if $(v_i, v_j) \in \mathcal{E}$, and $A_{ij} = 0$, otherwise. We write $|G|$ and $|\mathcal{E}|$ interchangeably to denote the number of edges of $G$. Throughout this paper, we use lower-case letters, such as $g, y, \mathbf{x}$, and $\mathbf{a}$, to represent realizations of the random objects $G, Y, \boldsymbol{X}$ and $\boldsymbol{A}$, respectively. Given a labeled graph $G = (\mathcal{V}, \mathcal{E}; Y, \boldsymbol{X}, \boldsymbol{A})$, we denote the corresponding graph without label as $\overline{G}$, and parameterize it by $(\mathcal{V}, \mathcal{E}; \boldsymbol{X}, \boldsymbol{A})$. The induced distribution of $\overline{G}$ is represented as $P_{\overline{G}}$, and its support by $\overline{\mathcal{G}}$.

**Graph and Node Classification Tasks:** In the classification tasks under consideration, we are given:

- A set of labeled training data $\mathcal{T} = \{(\overline{G}_i, Y_i) | Y_i \in \mathcal{Y}, i \in [|\mathcal{T}|]\}$, where $(\overline{G}_i, Y_i)$ corresponds to the $i$-th graph and its associated class label. The pairs $(\overline{G}_i, Y_i), i \in [|\mathcal{T}|]$ are generated independently according to an identical joint distribution induced by $P_{Y, \boldsymbol{X}, \boldsymbol{A}}$.
- A classification function (GNN model) $f(\cdot)$ trained to classify an unlabeled input graph $\overline{G}$ into its class $Y$. It takes $\overline{G}$ as input and outputs a probability distribution $P_Y$ on alphabet $\mathcal{Y}$. The reconstructed label $\widehat{Y}$ is produced randomly based on $P_Y$.

In node classification tasks, each graph $G_i$ denotes a $K$-hop sub-graph centered around node $v_i$, with a GNN model $f$ trained to predict the label for node $v_i$ based on the node representation of $v_i$ learned from $G_i$, whereas in graph classification tasks, $G_i$ is a random graph whose distribution is determined by the (general) joint distribution $P_{Y, \boldsymbol{A}, \mathbf{Z}}$, with the GNN model $f(\cdot)$ trained to predict the label for graph $G$ based on the learned representation of $\overline{G}$. Formally we define a classifier as follows.

**Definition 1** (**Classifier**). *For a classification task with underlying distribution $P_{Y, \boldsymbol{X}, \boldsymbol{A}}$, a classier is a function $f : \overline{\mathcal{G}} \to \Delta_{\mathcal{Y}}$. For a given $\epsilon > 0$, the classifier is called $\epsilon$-accurate if $P(\widehat{Y} \neq Y) \leq \epsilon$, where $\widehat{Y}$ is produced according to probability distribution $f(\overline{G})$.*

### 2.1 Information Theoretic Measures for Quantifying Explainability

In this section, we introduce two different but related notions of explainability, namely, explainability of a classification task, and explainability of a classifier for a given task. The interrelations between these two notions are quantified in the subsequent sections.

Given a classification task with underlying distribution $P_{Y, \boldsymbol{X}, \boldsymbol{A}}$, an explanation function for the task is a mapping $\Psi : \overline{G} \mapsto (\mathcal{V}_{exp}, \mathcal{E}_{exp})$ which takes an unlabeled graph $\overline{G} = (\mathcal{V}, \mathcal{E}; \boldsymbol{X}, \boldsymbol{A})$ as input and outputs a subset of nodes $\mathcal{V}_{exp} \subseteq \mathcal{V}$ and subset of edges $\mathcal{E}_{exp} \subseteq \mathcal{V}_{exp} \times \mathcal{V}_{exp}$. Loosely speaking, a *good* explanation $(\mathcal{V}_{exp}, \mathcal{E}_{exp})$ is a subgraph which is an (almost) sufficient statistic of $\overline{G}$ with respect to the true label $Y$, i.e., $I(Y; \overline{G} | \mathbb{1}_{\Psi(\overline{G})}) \approx 0$, where we have defined[2]:

$$I(Y; \overline{G} | \mathbb{1}_{\Psi(\overline{G})}) \triangleq \sum_{g_{exp}} P_{\Psi(\overline{G})}(g_{exp}) \sum_{y, \overline{g}} P_{Y, \overline{G}}(y, \overline{g} | g_{exp} \subseteq \overline{G}) \log \frac{P_{Y, \overline{G}}(y, \overline{g} | g_{exp} \subseteq \overline{G})}{P_Y(y | g_{exp} \subseteq \overline{G}) P_{\overline{G}}(\overline{g} | g_{exp} \subseteq \overline{G})}.$$

In practice, a desirable explanation function is one whose output size is significantly smaller than the original input size, i.e., $\mathbb{E}_G(|\Psi(\overline{G})|) \ll \mathbb{E}_G(|\overline{G}|)$. This is formalized below.

**Definition 2** (**Explainable Classification Task**). *Consider a classification task with underlying distribution $P_{Y, \boldsymbol{X}, \boldsymbol{A}}$. An explanation function for this task is a mapping $\Psi : \overline{\mathcal{G}} \to 2^{\mathcal{V}} \times 2^{\mathcal{E}}$. For a given pair of parameters $\kappa \in [0, 1]$ and $s \in \mathbb{N}$, the task is called $(s, \kappa)$-explainable if there exists an explanation function $\Psi : \overline{G} \mapsto (\mathcal{V}_{exp}, \mathcal{E}_{exp})$ such that:*

$$i) \quad I(Y; \overline{G} | \mathbb{1}_{\Psi(\overline{G})}) \leq \kappa, \qquad\qquad ii) \quad \mathbb{E}_G(|\mathcal{E}_{exp}|) \leq s.$$

A similar notion of explainability can be provided for a given classifier as follows.

**Definition 3** (**Explainable Classifier**). *Consider a classification task with underlying distribution $P_{Y, \boldsymbol{X}, \boldsymbol{A}}$ and a classier $f : \overline{\mathcal{G}} \to \Delta_{\mathcal{Y}}$. For a given pair of parameters $\zeta \in [0, 1]$ and $s \in \mathbb{N}$, the*

---

[2] $I(Y; \overline{G} | \mathbb{1}_{\Psi(\overline{G})})$ can be alternatively written as $\mathbb{E}_{\overline{G}'}(I(Y; \overline{G} | \Psi(\overline{G}') \subseteq \overline{G}))$, where $P_{Y \overline{G} \overline{G}'} = P_{Y \overline{G}} P_{\overline{G}'}$.

*classifier $f(\cdot)$ is called $(s, \zeta)$-explainable if there exists an explanation function $\Psi(\cdot)$ such that*

$$i) \quad I(\widehat{Y}; \overline{G}|\mathbb{1}_{\Psi(\overline{G})}) \leq \zeta, \qquad\qquad ii) \quad \mathbb{E}_G(|\mathcal{E}_{exp}|) \leq s,$$

*where $\Psi(\overline{G}) = (\mathcal{V}_{exp}, \mathcal{E}_{exp})$ and $\widehat{Y}$ is generated according to the probability distribution $f(\overline{G})$. The explanation function $\Psi(\cdot)$ is called an $(s, \zeta)$ explanation for $f(\cdot)$.*

## 3 FIDELITY MEASURES FOR EXPLAINABILITY

### 3.1 EXPLAINABLE TASKS VS EXPLIANABLE CLASSIFIERS

It should be noted that the explainability of the classification task (Definition 2) does not imply nor is it implied by the explainability of a classifier for that task (Definition 3). For instance, the trivial classifier whose output is independent of input is explainable for any task, even if the task is not explainable itself. In this section, we characterize a quantitative relation between these two notions of explainability. To keep the analysis tractable, we introduce the following condition on $\Psi(\cdot)$:

$$\text{Condition 1:} \quad \forall \overline{g}, \overline{g}' : \Psi(\overline{g}) \subseteq \overline{g}' \Rightarrow \Psi(\overline{g}') = \Psi(\overline{g}). \tag{1}$$

Condition 1 holds for the ground-truth explanation in many of the widely studied datasets in the explainability literature such as BA-2motifs, Tree-Cycles, Tree-Grid, and MUTAG datasets which are discussed in Section 5. An important consequence of Condition 1 is that if $\Psi(\cdot)$ satisfies the condition, then $I(\widehat{Y}; \overline{G}|\mathbb{1}_{\Psi(\overline{G})}) = I(\widehat{Y}; \overline{G}|\Psi(\overline{G}))$. In the following, under the assumption of Condition 1, we show that if the classifier has a low error probability, then its explainability implies the explainability of the underlying task. Conversely, we show that if the task is explainable, and its associated Bayes error rate is small, then any classifier for the task with an error close to the Bayes error rate is explainable. The following provides the main result of this section.

**Theorem 1 (Sufficient Conditions for Equivalency of Task and Classifier Explainability).** *Consider a classification task with underlying distribution $P_{Y, \mathbf{X}, \mathbf{A}}$, parameters $\zeta, \kappa, \epsilon \in [0, 1]$, and an integer $s \in \mathbb{N}$. Then,*

1. *If there exists a classifier $f(\cdot)$ for this task which is $\epsilon$-accurate and $(s, \zeta)$-explainable, with the explanation function satisfying Condition 1, then, the task is $(s, \kappa')$-explainable, where*

$$\kappa' \leq \zeta + h_b(\epsilon) + \epsilon \log (|\mathcal{Y}| - 1),$$

*and $h_b(p) \triangleq -p \log p - (1 - p) \log 1 - p, p \in [0, 1]$ denotes the binary entropy function. Particularly, $\kappa' \to 0$ as $\zeta, \epsilon \to 0$.*

2. *If the classification task is binary (i.e. $|\mathcal{Y}| = 2$), is $(s, \kappa)$-explainable with an explanation function satisfying Condition 1, and has Bayes error rate equal to $\epsilon^*$, then any $(\epsilon^* + \delta)$-accurate classifier $h(\cdot)$ is $(s, \eta)$-explainable, where*

$$\eta \triangleq h_b(\tau) + \tau, \quad \tau \triangleq \frac{\left(2\sqrt{2}\epsilon^* + \sqrt{\xi}\right)^2}{2}, \quad \xi \triangleq \delta + e_{max}(\epsilon^*, \kappa) - \epsilon^*,$$
$$\delta \in (0, 9\epsilon^* - e_{max}(\epsilon^*, \kappa)),$$

*and $e_{max}(\cdot)$ is defined in Proposition 1. Particularly, $\eta \to 0$ as $\delta, \epsilon^*, \kappa \to 0$.*

The proof of Part 1) follows from the definition of explainability in Definitions 2 and 3, and Fano's inequality. The proof of Part 2) uses the following intermediate result, which proves the existence of an explainable classifier for any explainable task, whose error is close to that of the Bayes classifier.

**Proposition 1 (Existence of Accurate and Explainable Classifiers for Explainable Tasks).** *Consider a classification task with underlying distribution $P_{Y, \mathbf{X}, \mathbf{A}}$, parameters $\kappa, \epsilon \in [0, 1]$, and an integer $s \in \mathbb{N}$. Assume that the task is $(s, \kappa)$-explainable with an exaplantion function satisfying Condition 1. Further assume that the classification task has a Bayes error rate equal to $\epsilon^*$. Then, there exists an $\epsilon$-accurate and $(s, 0)$-explainable classifier $f(\cdot)$, such that,*

$$\epsilon \leq e_{max}(\epsilon^*, \kappa), \tag{2}$$

*where we have defined*

$$e_{max}(x, y) \triangleq \arg\max_z \{z | e_{RF}(z) \leq h_b(x) + x \log(|\mathcal{Y}| - 1) + y\}, \quad x, y \in [0, 1]$$

$$e_{RF}(z) \triangleq \left(1 - (1-z)\left\lfloor \frac{1}{1-z} \right\rfloor\right)\left(1 + \left\lfloor \frac{1}{1-z} \right\rfloor\right)\ln\left(1 + \left\lfloor \frac{1}{1-z} \right\rfloor\right)$$
$$- \left(z - (1-z)\left\lfloor \frac{1}{1-z} \right\rfloor\right)\left\lfloor \frac{1}{1-z} \right\rfloor \ln\left\lfloor \frac{1}{1-z} \right\rfloor, \quad z \in (0, 1).$$

*In particular, $\epsilon \to 0$ as $\epsilon^*, \kappa \to 0$.*

The complete proof of Proposition 1 is provided in Appendix A.1, and the proof of Theorem 1 is provided in Appendix A.2.

### 3.2 SURROGATE FIDELITY MEASURES FOR QUANTIFYING EXPLAINABILITY

Definitions 2 and 3 provide intuitive notions of explainability along with fidelity measures expressed as mutual information terms, however, in most practical applications it is not possible to quantitatively compute and analyze them. To elaborate, let us consider the mutual information term in condition i) of Definition 3. Estimating the mutual information term, $I(\widehat{Y}; \overline{G}|\mathbb{1}_{\Psi(\overline{G})})$ is not practically feasible in most applications since $\overline{G}$ has large alphabet size. To address this, prior works have considered alternative *surrogate* fidelity measures for evaluating the performance of explanation functions. At a high level, an ideal surrogate measure $Fid^* : (f, \Psi) \mapsto \mathbb{R}^{\geq 0}$ must satisfy two properties: i) The surrogate fidelity value $Fid^*(f, \Psi)$ must be monotonic with the mutual information term $I(\widehat{Y}; \overline{G}|\mathbb{1}_{\Psi(\overline{G})})$ given in Definition 3, so that a 'good' explanation function with respect to the surrogate fidelity measure is 'good' under Definition 3 and vice versa, and ii) there must exist an empirical estimate of $Fid^*(f, \Psi)$ with sufficiently fast convergence guarantees so that the surrogate measure can be estimated accurately using a reasonably large set of observations. These conditions are formalized in the following two definitions.

**Definition 4 (Surrogate Fidelity Measure).** *For a classification task with underlying distribution $P_{Y, \mathbf{X}, \mathbf{A}}$, a (surrogate) fidelity measure is a mapping $Fid : (f, \Psi) \to \mathbb{R}^{\geq 0}$, which takes a pair consisting of a classification function $f(\cdot)$ and explanation function $\Psi(\cdot)$ as input, and outputs a non-negative number. The fidelity measure is said to be well-behaved for a set of classifiers $\mathcal{F}$ and explanation functions $\mathcal{S}$ if for all pairs of explanation functions $\Psi_1, \Psi_2 \in \mathcal{S}$ and classifiers $f \in \mathcal{F}$, we have:*

$$I(\widehat{Y}; \overline{G}|\mathbb{1}_{\Psi_1(\overline{G})}) \leq I(\widehat{Y}; \overline{G}|\mathbb{1}_{\Psi_2(\overline{G})}) \iff Fid(f, \Psi_2) \leq Fid(f, \Psi_1). \quad (3)$$

The condition in equation 3 requires that better explanation functions in the sense of Definition 3 must have higher fidelity when evaluated using the surrogate measure.

**Definition 5 (Rate of Convergence Guarantee).** *Let $\mathcal{T}_i = \{(\overline{G}_j, Y_j)|j \leq i\}, i \in \mathbb{N}$ be a sequence of sets of independent and identically distributed observations for a given classification problem. A fidelity measure $Fid(\cdot, \cdot)$ is said to be empirically estimated with rate of convergence $\beta$ if there exits a sequence of functions $H_n : \mathcal{T}_n \mapsto \widehat{Fid}_n, n \in \mathbb{N}$ such that for all $\epsilon > 0$, we have:*

$$P(|Fid(f, \Psi) - \widehat{Fid}_n(f, \Psi)| > \epsilon) = O(n^{-\beta}),$$

*for all classifiers $f$ and explanation functions $\Psi$.*

#### 3.2.1 THE OOD PROBLEM IN EVALUATING SURROGATE FIDELITY MEASURES

One class of surrogate fidelity measures has been considered in several recent works (e.g., (Pope et al., 2019; Yuan et al., 2022)) are the $Fid_+$, $Fid_-$, and $Fid_\Delta$ measures:

$$Fid_+ \triangleq \mathbb{E}(\widehat{P}(Y) - \widehat{P}^+(Y)), \quad Fid_- \triangleq \mathbb{E}(\widehat{P}(Y) - \widehat{P}^-(Y)), \quad Fid_\Delta \triangleq Fid_+ - Fid_-, \quad (4)$$

where $\widehat{P}(\cdot)$ is the distribution given by $f(\overline{G})$, $\widehat{P}^+(\cdot)$ is the distribution given by $f(\overline{G}_i - \Psi(\overline{G}_i))$, $\widehat{P}^-(\cdot)$ is the distribution given by $f(\Psi(\overline{G}_i))$, $\overline{G} - \Psi(\overline{G})$ is the subgraph with edge set $\mathcal{E} - \mathcal{E}_{exp}$ for

$\Psi(\overline{G}) = (\mathcal{V}_{exp}, \mathcal{E}_{exp})$, and $\mathcal{T}$ is a set of independent observations $\mathcal{T} = \{(\overline{G}_i, Y_i), i = 1, 2, \cdots, |\mathcal{T}|\}$. As mentioned in (Yuan et al., 2022), these fidelity measures can be empirically estimated by

$$\widehat{Fid}_+ \triangleq \frac{1}{|\mathcal{T}|}\sum_{i=1}^{|\mathcal{T}|}\widehat{P}(y_i) - \widehat{P}^+(y_i), \quad \widehat{Fid}_- \triangleq \frac{1}{|\mathcal{T}|}\sum_{i=1}^{|\mathcal{T}|}\widehat{P}(y_i) - \widehat{P}^-(y_i), \quad \widehat{Fid}_\Delta \triangleq \widehat{Fid}_+ - \widehat{Fid}_-.$$

It should be noted that the rate of convergence of this empirical estimate is $\beta = \frac{1}{2}$ (e.g., using the Berry- Esseen theorem (Billingsley, 2017)). The following proposition shows that these measures are well-behaved for a class of deterministic classification tasks and completely explainable classifiers.

**Proposition 2 (Well-Behavedness of $Fid_\Delta$ on Deterministic Tasks).** *Consider a deterministic classification task for which the induced distribution $P_{\overline{G}}$ has support $\overline{\mathcal{G}}$ consisting of all graphs with $n \in \mathbb{N}$ vertices, the graph edges are jointly independent, and $\boldsymbol{X} \in \mathcal{Z}^{n \times d}$, where $\mathcal{Z}$ is a finite set. Further assume that the graph label is $Y = \mathbb{1}(g_{exp} \subseteq \overline{G})$ for a fixed subgraph $g_{exp}$, so that the task is deterministic. Let $f(\overline{G}) = \mathbb{1}(g_{exp} \subseteq \overline{G})$ be the 0-correct classifier. Let $\mathcal{S} = \{\Psi_p(\overline{g}|p \in [0,1]\}$ be a class of explanation functions, where $P(\Psi_p(\overline{g}) = \overline{g}_{epx}|\overline{g}_{exp} \in \overline{g}) = p$ and $P(\Psi_p(\overline{g}) = \phi|\overline{g}_{exp} \in \overline{g}) = 1 - p, p \in [0,1]$. The $Fid_\Delta$ fidelity measure is well-behaved for all explanation functions in $\mathcal{S}$.*

The proof is provided in Appendix A.3. Proposition 2 shows that $Fid_\Delta$ is well-behaved in a specific set of scenarios, where the task is deterministic and the classifier is completely explainable. However, we argue that it is not well-behaved in a wide range of scenarios of interest which do not have these properties. This is due to the OOD issue mentioned in the introduction. To elaborate, for a good classifier, which has low probability of error, the distribution $f(\overline{G})$ must be close to $P_{Y|\overline{G}}(\cdot|\overline{G})$ on average, i.e., $\mathbb{E}_{\overline{G}}(d_{TV}(f(\overline{G}), P_{Y|\overline{G}}(\cdot|\overline{G})))$ should be small, where $d_{TV}$ denotes the total variation. As a result, $\widehat{P}(Y)$ in equation 4 is close to $P_{Y|\overline{G}}(Y|\overline{G})$ on average. However, this is not necessarily true for the $\widehat{P}^+(Y)$ and $\widehat{P}^-(Y)$ terms. The reason is that the assumption $\mathbb{E}_{\overline{G}}(d_{TV}(f(\overline{G}), P_{Y|\overline{G}}(\cdot|\overline{G}))) \approx 0$ only ensures that $f(\overline{G})$ is close to $P_{Y|\overline{G}}(\cdot|\overline{G})$ for the *typical* realizations of $\overline{G}$. However, $\overline{G} - \Psi(\overline{G})$ and $\Psi(\overline{G})$ are not typical realizations. For instance, in many applications, it is very unlikely or impossible to observe the explanation graph in isolation, that is, to have $\overline{G} = \Psi(\overline{G})$. As a result, $\widehat{P}^+(Y)$ and $\widehat{P}^-(Y)$ are not good approximations for $P_{Y|\overline{G}}(Y|\overline{G} - \Psi(\overline{G}))$ and $P_{Y|\overline{G}}(Y|\Psi(\overline{G}))$, respectively, and $Fid_+, Fid_-$ and $Fid_\Delta$ are not well-behaved. This is observed in our empirical analysis in Section 5, and the notion is analytically investigated in a toy-example in Appendix B.

### 3.2.2 A Robust Class of Surrogate Fidelity Measures

Generally, in scenarios where $\Psi(\overline{G})$ and $\overline{G} - \Psi(\overline{G})$ are not typical with respect to the distribution of $\overline{G}$, the $Fid_\Delta$ measure may not be well-behaved. We address this by introducing a class of modified fidelity measures by modifying the definitions of $Fid_+$ and $Fid_-$ in equation 4. To this end, we define the stochastic graph sampling function $E_\alpha : \overline{G} \mapsto \overline{G}_\alpha$ with edge erasure probability $\alpha \in [0,1]$. That is, $E_\alpha(\cdot)$ takes a graph $\overline{G}$ as input, and outputs a sampled graph $\overline{G}_\alpha$ whose vertex set is the same as that of $\overline{G}$, and its edges are sampled from $\overline{G}$ such that each edge is included with probability $\alpha$ and erased with probability $1 - \alpha$, independently of all other edges. We introduce the following generalized class of surrogate fidelity measures, and show that they are robust to OOD issues in a wide range of scenarios:

$$Fid_{\alpha_1,+} \triangleq \mathbb{E}(\widehat{P}(Y) - \widehat{P}^{\alpha_1,+}(Y)), \qquad Fid_{\alpha_2,-} \triangleq \mathbb{E}(\widehat{P}(Y) - \widehat{P}^{\alpha_2,-}(Y)), \qquad (5)$$

$$Fid_{\alpha_1,\alpha_2,\Delta} \triangleq Fid_{\alpha_1,+} - Fid_{\alpha_2,-}, \qquad (6)$$

where $\alpha_1, \alpha_2 \in [0,1]$, $\widehat{P}(\cdot)$ is the distribution given by $f(\overline{G})$, $\widehat{P}^{\alpha_1,+}(\cdot)$ is the distribution given by $f(\overline{G} - E_{\alpha_1}(\Psi(\overline{G})))$, $\widehat{P}^{\alpha_2,-}(\cdot)$ is the distribution given by $f(E_{\alpha_2}(\overline{G} - \Psi(\overline{G})) + \Psi(\overline{G}))$.

Note that if $\alpha_1 = 1$ and $\alpha_2 = 0$, we recover the original fidelity measures, i.e., $Fid_{1,+} = Fid_+$, $Fid_{0,-} = Fid_-$, and $Fid_{1,0\Delta} = Fid_\Delta$. On the other hand, if $\alpha_1 = 0$ and $\alpha_2 = 1$, we have $\overline{G} - E_{\alpha_1}(\Psi(\overline{G})) = E_{\alpha_2}(\overline{G} - \Psi(\overline{G})) + \Psi(\overline{G}) = \overline{G}$. Consequently, in this case there would be no OOD issue, however, the resulting fidelity measure is not informative since $Fid_{0,1,\Delta}(f, \Psi) = 0$, for all classifiers $f$ and explanation functions $\Psi$. Loosely speaking, if $P_{Y|\overline{G}}(y|\overline{g}), y \in \mathcal{Y}$ is 'smooth' as a

function of $\overline{g}$, then OOD does not manifest and $\alpha_1 \approx 1, \alpha_2 \approx 0$ yields a suitable fidelity measure (e.g., Proposition 2). On the other hand if $P_{Y|\overline{G}}(y|\overline{g}), y \in \mathcal{Y}$ is not smooth, then $\alpha_1 < 1$ and $\alpha_2 > 0$ would yield a suitable fidelity measure as this choice avoids the OOD problem. The generalized fidelity measures can be empirically estimated by

$$\widehat{Fid}_{\alpha_1,\epsilon,+} \triangleq \frac{1}{|\mathcal{T}|} \sum_{i=1}^{|\mathcal{T}|} \frac{1}{|\mathcal{A}^\epsilon_{|\mathcal{E}_{exp}|}(\alpha_1)|} \sum_{k_1 \in \mathcal{B}_{|\mathcal{E}_{exp}|,\alpha_1,\epsilon}} \sum_{\substack{\mathcal{E} \subseteq \mathcal{E}_{exp}: \\ |\mathcal{E}|=k_1}} \widehat{P}_{\mathcal{E}_i}(y_i) - \widehat{P}_{\mathcal{E}_i - \mathcal{E}}(y_i)$$

$$\widehat{Fid}_{\alpha_2,\epsilon,-} \triangleq \frac{1}{|\mathcal{T}|} \sum_{i=1}^{|\mathcal{T}|} \frac{1}{|\mathcal{A}^\epsilon_{|\mathcal{E}_i|}(\alpha_2)|} \sum_{k_2 \in \mathcal{B}_{|\mathcal{E}_i|,\alpha_2,\epsilon}} \sum_{\substack{\mathcal{E} \subseteq \mathcal{E}_i: \\ |\mathcal{E}|=k_2}} \widehat{P}_{\mathcal{E}_i}(y_i) - \widehat{P}_{\mathcal{E} \cup \mathcal{E}_{exp}}(y_i)$$

$$\widehat{Fid}_{\alpha_1,\alpha_2,\epsilon,\Delta} \triangleq \widehat{Fid}_{\alpha_1,\epsilon,+} - \widehat{Fid}_{\alpha_2,\epsilon,+},$$

where $\epsilon > 0$, $\mathcal{B}_{\ell,\alpha,\epsilon}, \ell \in \mathbb{N}, \alpha, \epsilon > 0$ denotes the interval $[\ell(\alpha - \epsilon), \ell(\alpha + \epsilon)]$, the set $\mathcal{A}^\epsilon_\ell(\alpha), \ell \in \mathbb{N}, \alpha \in [0,1], \epsilon > 0$ is the set of $\epsilon$-typical binary sequences of length $\ell$ with respect to the Bernoulli distribution with parameter $\alpha$, i.e., $\mathcal{A}^\epsilon_\ell(\alpha) \triangleq \{x^\ell \in \{0,1\}^\ell | |\frac{1}{\ell} \sum_{i=1}^\ell \mathbb{1}(x_i = 1) - \alpha| \leq \epsilon\}$, the distribution $\widehat{P}_{\mathcal{E}}(y_i)$ is the probability of $y_i$ under the distribution $f(\overline{G}_\mathcal{E})$, where $\overline{G}_\mathcal{E}$ is the subgraph of $\overline{G}$ with edge set restricted to $\mathcal{E}$, and $\mathcal{T} = \{(\overline{G}_i, Y_i)|i \in [|\mathcal{T}|]\}$ is the set of observations, where $\mathcal{E}_i$ is the edge set of $\mathcal{G}_i, i \in [\mathcal{T}]$. Using the Chernoff bound and standard information theoretic arguments, it can be shown that for fixed $\alpha_1, \alpha_2$ and $\epsilon = O(\sqrt{\frac{1}{|\mathcal{E}|}})$, these empirical estimates converge to their statistical counterparts with rate of convergence $\beta = \frac{1}{2}$ as $|\mathcal{T}| \to \infty$ for large input graph and explanation sizes (e.g., (Csiszár & Körner, 2011)).

We show that $Fid_{\alpha_1,\alpha_2,\Delta}$ is well-behaved for a general class of tasks and classifiers, where the original Fidelity measure, $Fid_{1,0,\Delta}$ is not well-behaved. Specifically, we assume that there exists a set of *motifs* $\overline{g}_y, y \in \mathcal{Y}$, such that

$$P(Y = y|\overline{G} = \overline{g}) = \begin{cases} 1 & \text{if} \quad \overline{g}_y \subseteq \overline{g} \text{ and } \forall y' \neq y : \overline{g}_{y'} \nsubseteq \overline{g}, \\ 0 & \text{if } \exists! y' \neq y : \overline{g}_y \nsubseteq \overline{G}, \overline{g}_{y'} \subseteq \overline{G} \\ \frac{1}{|\mathcal{Y}|} & \text{otherwise.} \end{cases}$$

Furthermore, given $n \in \mathbb{N}$ and $\delta, \epsilon \in [0,1]$, we assume that the graph distribution $P_{\overline{G}}$ and the trained classifier $f_\delta(\cdot)$, satisfy the following conditions. The graph has $n$ vertices. There exists a set $\mathcal{G}$ of input graphs, called an $\epsilon$-typical set, such that $P_{\overline{G}}(\mathcal{G}) > 1 - \epsilon$, and

$$P(f_\delta(\overline{g}) = f^*(\overline{g})) = (\frac{1}{d(\mathcal{G},\overline{g}) + 1})^\delta, \tag{7}$$

where $f^*(\cdot)$ is the optimal Bayes classifier, i.e., $f^*(\overline{g}) = y$ if $\overline{g}_y \subseteq \overline{g}$, and $f^*(\overline{g}) = \arg\max_y P_Y(y)$, otherwise.[3] The distance between the graph $\overline{g}$ and the set of graphs $\mathcal{G}$ is defined as $d(\mathcal{G},\overline{g}) \triangleq \min_{\overline{g}' \in \mathcal{G}} d(\overline{g}',\overline{g})$, and the distance between two graphs is defined as their number of edge differences.

**Proposition 3 (Well-Behavedness of $Fid_{\alpha_1,\alpha_2,\Delta}$ Fidelity Measure).** *In the classification scenario described above, Consider the class of explanation functions $\mathcal{S}$, which consists of stochastic mappings $\Psi : \overline{g} \mapsto \overline{G}_{exp}$ where $P(\overline{G}_{exp} = \overline{g}_y|\overline{G} = \overline{g}) = p$ for any $\overline{g}$ such that $\exists! y : \overline{g}_y \in \overline{g}$ and $\overline{G}_{exp} = \phi$, otherwise. Then,*

$$Fid_{\alpha_1,+} \geq \epsilon^* - \epsilon - \frac{P(\mathcal{A})}{1 - \epsilon}((1 - p + p\max_y P_Y(y)) + 1 - (\frac{1}{k+1})^\delta) - P(\mathcal{A}^c) - 1 - P(\mathcal{B}') - \epsilon$$

$$Fid_{\alpha_2,-} \leq \epsilon^* - P(\mathcal{A})(1 - 2^{\frac{k^2}{2n^2}})(\frac{1}{k+1})^\delta p$$

$$Fid_{\alpha_1,\alpha_2,\Delta} \geq -\epsilon - P(\mathcal{A}^c) - 1 - P(\mathcal{B}') - \frac{P(\mathcal{A})}{1 - \epsilon}((1 - p + p\max_y P_Y(y)) +$$

$$1 - \left(\frac{1}{k+1}\right)^\delta) + P(\mathcal{A})\left(1 - 2^{\frac{k^2}{2n^2}}\right)\left(\frac{1}{k+1}\right)^\delta p.$$

---

[3]The assumption in equation 7 captures the infrequency of occurrence of atypical inputs in the training set, which increases chance of missclassifcation for those inputs and hence gives rise to OOD issues.

*where $\epsilon^*$ is the Bayes error rate, $\alpha_1 \triangleq \frac{k}{2n^2}$, $\alpha_2 \triangleq 1 - \frac{k}{2n^2}$, $k < s_1 \triangleq \min_y |\overline{g}_y|$, $\mathcal{A}$ is the event that $\exists! y : \overline{g}_y \subseteq \overline{G}$, and $\mathcal{B}'$ is the event that $\sum_{i=1}^{n^2} X_i > k$, where $X_i$ are independent and identically distributed realizations of a Bernoulli variable with parameter $\alpha_1$. Particularly, as $k, n \to \infty$ and $\delta, \epsilon \to 0$ such that $k = o(n)$ and $\delta = o(\frac{1}{k})$, $Fid_{\alpha_1, \alpha_2, \Delta}$ becomes monotonically increasing in $p$. Consequently, there exists a non-zero error threshold $\epsilon_{th} > 0$, such that $Fid_{\alpha_1, \alpha_2, \Delta}$ is well-behaved for all $\epsilon^* \le \epsilon_{th}$.*

The proof is provided in Appendix A.4

## 4 RELATED WORK

**Explainable GNNs.** To interpret GNN models, previous works (Ying et al., 2019; Luo et al., 2020; Yuan et al., 2020; 2022; 2021; Lin et al., 2021; Wang & Shen, 2023; Miao et al., 2023; Fang et al., 2023c; Xie et al., 2022; Ma et al., 2022) tried different methods to offer interpretability. According to granularity, these explanation methods can generally be divided into two categories: i) instance-level explanation (Ying et al., 2019; Zhang et al., 2022b; Xie et al., 2022), which offers explanations for every instance by recognizing important substructures; and ii) model-level explanation (Yuan et al., 2020; Wang & Shen, 2023; Azzolin et al., 2023), which is designed for providing global decision rules learned by target GNN models. Within these methodologies, these methods can be classified as post-hoc explanations (Ying et al., 2019; Luo et al., 2020; Yuan et al., 2021) and self-explainable GNNs (Baldassarre & Azizpour, 2019; Dai & Wang, 2021; Miao et al., 2022), where the former uses an extra GNN model to elucidate the target GNN and the latter offers explanations while making predictions. For a comprehensive survey on this topic, please refer to (Yuan et al., 2022).

**Evaluation Metrics for Explainable GNNs.** To comprehensively evaluate the explanations, both anecdotal evidence and quantitative measures, such as explanation accuracy, faithfulness, efficiency, completeness, consistency, are jointly adopted in the literature (Nauta et al., 2022). When ground-truth explanations are unavailable, fidelity measures and their variants are routinely adopted to evaluate the quality of explanations (Yuan et al., 2021). For example, DIG (Liu et al., 2021) and GraphFramEx (Amara et al., 2022) include both $Fid_+$ and $Fid_-$; GraphXAI (Agarwal et al., 2023) adopts a variant that utilizes Kullback-Leibler (KL) divergence score to quantify the distance between predictions of the original graph and subgraph. GNNX-BENCH (Kosan et al., 2023) uses Fidelity to evaluate both factual and counterfactual explanations. However, the OOD problem of subgraph explanations is overlooked in these platforms. Concurrently to this work, OAR evaluates post-hoc explanation subgraphs via analyzing adversarial robustness (Fang et al., 2023a); GInX-Eval evaluates with a removing and fine-tuning strategy (Amara et al., 2023).

## 5 EXPERIMENTS

In this section, we empirically verify the effectiveness of the generalized class of surrogate fidelity measures. We also conduct extensive studies to verify our theoretical claims. Four benchmark datasets with ground truth explanations are used for evaluation, with Tree-Circles, and Tree-Grid (Ying et al., 2019) for the node classification task, and BA-2motifs (Luo et al., 2020) and MUTAG (Debnath et al., 1991) for graph classification. We consider both GCN and GIN architectures (Ying et al., 2019; Xu et al., 2019) as the models to be explained. Detailed experimental setups, full experimental results, and extra experiments are presented in the Appendix.

### 5.1 QUANTITATIVE EVALUATION BY COMPARING TO THE GOLD STANDARD

In adopted datasets, motifs are included which determine node labels or graph labels. Thus, the relationships between graph examples and data labels are well-defined by humans. The correctness of an explanation can be evaluated by comparing it to the ground truth motif. Previous studies (Ying et al., 2019; Luo et al., 2020) usually model the evaluation as an edge classification problem. Specifically, edges in the ground-truth explanation are treated as labels, and importance weights given by the explainability method are viewed as prediction scores. Then, AUC scores are considered as the metric for correctness. In this section, we use a more tractable metric, edit distance (Gao et al., 2010) to compare achieved explanations with the ground-truth motifs as a Gold Standard metric.

Consider an input graph $\overline{G}_i$ and let $\overline{G}_{i,exp}$ be the ground-truth explanation subgraph, i.e., the motif. We construct a set of explanation functions, with varying qualities, to evaluate the well-behavedness

Table 1: Spearman correlation coefficient between metric and gold standard edit distance.

| | Dataset | $Fid_+ \downarrow$ | $Fid_{\alpha_1,+} \downarrow$ | $Fid_- \uparrow$ | $Fid_{\alpha_2,-} \uparrow$ | $Fid_\Delta \downarrow$ | $Fid_{\alpha_1,\alpha_2,\Delta} \downarrow$ | AUC |
|---|---|---|---|---|---|---|---|---|
| GCN | Tree-Cycles | 0.229 | **-0.990** | -0.210 | **0.990** | 0.105 | **-0.990** | -1.000 |
| | Tree-Grid | 0.095 | **-1.000** | 0.457 | **1.000** | -0.781 | **-1.000** | -1.000 |
| | BA-2motifs | -0.924 | **-1.000** | 0.819 | **1.000** | -0.990 | **-1.000** | -1.000 |
| | MUTAG | -0.190 | **-1.000** | -0.276 | **1.000** | -0.105 | **-1.000** | -1.000 |
| GIN | Tree-Cycles | 0.200 | **-1.000** | -0.229 | **1.000** | -0.286 | **-1.000** | -1.000 |
| | Tree-Grid | 1.000 | 1.000 | -1.000 | -0.993 | 1.000 | 1.000 | -1.000 |
| | BA-2motifs | -0.838 | **-1.000** | 0.905 | **1.000** | **-1.000** | **-1.000** | -1.000 |
| | MUTAG | **-1.000** | **-1.000** | 0.886 | **1.000** | -0.990 | **-1.000** | -1.000 |

of the proposed fidelity measures. To elaborate, for a given $\beta_1, \beta_2 \in [0,1]$, we construct an explanation function $\Psi_{\beta_1,\beta_2}(\cdot)$ by random IID sampling of the explanation subgraph edges, and the non-explanation subgraph edges, with sampling rates $\beta_1$ and $\beta_2$, respectively. That is, to construct $\Psi_{\beta_1,\beta_2}(\overline{G}_i)$, we remove $\beta_1$ ratio of edges from ground-truth explanation $\overline{G}_{i,exp}$ via random IID sampling, and randomly add $\beta_2 \in [0,1]$ ratio of edges from $\overline{G}_i - \overline{G}_{i,exp}$ to $\overline{G}_{i,exp}$ by random IID sampling from the non-explanation subgraph. Clearly, the explanation function should receive a better fidelity score for smaller $(\beta_1, \beta_2)$. We sweep $\beta_1$ and $\beta_2$ in the range $[0, 0.1, 0.3, 0.5, 0.7, 0.9]$, and for each combination $(\beta_1, \beta_2)$, we randomly sample 10 candidate explanations. We adopt the proposed $Fid_{\alpha_1,+}, Fid_{\alpha_2,-}, Fid_{\alpha_1,\alpha_2,\Delta}$, where we have taken $\alpha_1 = 1 - \alpha_2 = 0.1$, as well as their counterparts to evaluate their qualities. For each combination, we calculate the average metric scores.

As analyzed in previous works (Yuan et al., 2021), fidelity measurements ignore the size of the explanation. Thus, redundant explanations are usually with high $Fid_+$ and low $Fid_-$ scores. In the extreme case, with the whole input graph as the explanation, fidelity measures achieve the trivial optimal scores. This limitation is inherent and cannot not solved with the proposed metrics. To fairly compare the proposed metrics with the original ones, for each $\beta_2$, given a fidelity measurement, we use the Spearman correlation coefficient (Myers et al., 2013) between it and the gold standard edit distance to quantitatively evaluate the quality of the metric. Then, we report the average correlation scores in Table 1. The number of sampling in our measurements are set to 50.

We have the following observations in Table 1. First, $Fid_{\alpha_1,+}$ consistently yields correlation scores near -1.0 with the edit distance. It signifies a robust inverse relationship between the two metrics. This is in agreement with the requirement in equation 3. In contrast, the original $Fid_+$ metric exhibits mixed results with half-positive and half-negative correlations. This inconsistency in $Fid_+$ underscores the potential superiority and consistency of our proposed $Fid_{\alpha_1,+}$ in aligning closely with the edit distance across various datasets. Moreover, the proposed $Fid_{\alpha_2,-}$ is strongly positively related to gold-standard edit distance compared to the original $Fid_-$. we have similar observations in $Fid_{\alpha_1,\alpha_2,\Delta}$ and $Fid_\Delta$. Third, we observe that the AUC score of edge classification, which is used in previous papers, is perfectly aligned with the gold standard edit distance, which verifies the correctness of using AUC as the metric. Last, all fidelity measurements fail to evaluate the GIN classifier on the Tree-Grid dataset. The potential reason is that GIN was designed for graph classification and its generalization performance on node classification is limited in Tree-Grid dataset.

## 6 CONCLUSION

In this paper, we have comprehensively explored the limitations intrinsic to prevalent evaluation methodologies in the realm of explainable graph learning, emphasizing the pitfalls of conventional fidelity metrics, notably their vulnerability to distribution shifts. By delving deep into the theoretical aspects, we have innovated a novel set of evaluation metrics grounded in information theory, offering resilience to such shifts and promising enhanced authenticity and applicability in real-world scenarios. These proposed metrics have been rigorously validated across varied datasets, demonstrating their superior alignment with gold standard benchmarks. Our endeavor contributes to the establishment of a more robust and reliable machine-learning evaluation framework for graph learning.

## ACKNOWLEDGMENTS

This project was partially supported by NSF grants IIS-2331908 and CCF-2241057. The views and conclusions contained in this paper are those of the authors and should not be interpreted as representing any funding agencies.

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

## A PROOFS

### A.1 PROOF OF PROPOSITION 1

*Proof.* Let $\Psi(\cdot)$ be the explanation corresponding to the $(s, \kappa)$-explainable task, and let $f^*(\overline{g}) \triangleq \arg\max_{y \in \mathcal{Y}} P_{Y|\overline{G}}(y|\overline{g})$ be the Bayes classifier. Define $f(\overline{g})$ as the deterministic classifier which outputs $\arg\max_{y \in \mathcal{Y}} P_{Y|\Psi(\overline{G})}(y|\Psi(\overline{g}))$ for a given input $\overline{g}$. By construction, $f(\cdot)$ is $(s, 0)$-explainable since $I(\widehat{Y}; \overline{G}|\mathbb{1}_{\Psi(\overline{G})}) = 0$. It remains to show that equation 2 holds, where $\epsilon$ is the accuracy of $f(\cdot)$ and $\epsilon^*$ is the accuracy of $f^*(\cdot)$. To this end, note that by the assumption of explainability of the task, and using the fact that under Condition 1 we have $I(Y; \overline{G}|\mathbb{1}_{\Psi(\overline{G})}) = I(Y; \overline{G}|\Psi(\overline{G}))$, it follows that:

$$I(Y; \overline{G}|\mathbb{1}_{\Psi(\overline{G})}) = I(Y; \overline{G}|\Psi(\overline{G})) \leq \kappa \Rightarrow H(Y|\Psi(\overline{G})) \leq H(Y|\overline{G}) + \kappa.$$

By Fano's inequality, we have:

$$H(Y|\overline{G}) \leq h_b(\epsilon^*) + \epsilon^* \log(|\mathcal{Y}| - 1) \Rightarrow H(Y|\Psi(\overline{G})) \leq h_b(\epsilon^*) + \epsilon^* \log(|\mathcal{Y}| - 1) + \kappa.$$

On the other hand, by reverse Fano's inequality (Tebbe & Dwyer, 1968; Kovalevsky, 1968; Sakai & Iwata, 2017), we have:

$$H(Y|\Psi(\overline{G})) \geq \left(1 - (1 - \epsilon)\left\lfloor\frac{1}{1-\epsilon}\right\rfloor\right)\left(1 + \left\lfloor\frac{1}{1-\epsilon}\right\rfloor\right)\ln\left(1 + \left\lfloor\frac{1}{1-\epsilon}\right\rfloor\right)$$
$$- \left(\epsilon - (1 - \epsilon)\left\lfloor\frac{1}{1-\epsilon}\right\rfloor\right)\left\lfloor\frac{1}{1-\epsilon}\right\rfloor\ln\left\lfloor\frac{1}{1-\epsilon}\right\rfloor.$$

Consequently,

$$e_{RF}(\epsilon) \leq h_b(\epsilon^*) + \epsilon^* \log(|\mathcal{Y}| - 1) + \kappa.$$

This completes the proof. □

### A.2 PROOF OF THEOREM 1

*Proof.* Proof of 1): From the assumption that $f(\cdot)$ is $(s, \zeta)$-explainable, we conclude that there exists an explanation $\Psi(\cdot)$ satisfying conditions i) and ii) in Definition 3 and Condition 1 in equation 1. Then,

$$I(Y; \overline{G}|\mathbb{1}_{\Psi(\overline{G})}) = I(Y; \overline{G}|\Psi(\overline{G})) \leq I(Y, \widehat{Y}; \overline{G}|\Psi(\overline{G}))$$
$$\overset{(a)}{=} I(\widehat{Y}; \overline{G}|\Psi(\overline{G})) + I(Y; \overline{G}|\widehat{Y}, \Psi(\overline{G}))$$
$$\overset{(b)}{\leq} \zeta + I(Y; \overline{G}|\widehat{Y}, \Psi(\overline{G}))$$
$$\overset{(c)}{\leq} \zeta + H(Y|\widehat{Y}, \Psi(\overline{G}))$$
$$\overset{(d)}{\leq} \zeta + H(Y|\widehat{Y})$$
$$\overset{(e)}{\leq} \zeta + h_b(\epsilon) + \epsilon \log(|\mathcal{Y} - 1|),$$

where (a) follows from the chain rule of mutual information, (b) follows from the assumption that $\Psi(\cdot)$ is an $(s, \zeta)$ explanation for $f(\cdot)$, (c) follows from the information theoretic identity that $I(X; Y) \leq H(X)$ for all random variables $X, Y$, (d) follows from the fact that conditioning reduces entropy, and (e) follows from Fano's inequality (Cover, 1999).

Proof of 2): Following the proof of Proposition 1, define $f(\overline{g})$ as the deterministic classifier which outputs $\arg\max_{y \in \mathcal{Y}} P_{Y|\Psi(\overline{G})}(y|\Psi(\overline{g}))$ for a given input $\overline{g}$. We have:

$$P(h(\overline{G}) \neq f(\overline{G})) \leq P(h(\overline{G}) \neq f^*(\overline{G})) + P(f(\overline{G}) \neq f^*(\overline{G})) \tag{8}$$

Define $\mathcal{G}_\gamma \triangleq \{\overline{g} | |2P_{Y|\overline{G}}(1|\overline{g}) - 1| \leq 1 - \gamma\}$, where $\gamma \in [0, 1]$. We have,

$$P(h(\overline{G}) \neq f^*(\overline{G})) \leq P(\overline{G} \in \mathcal{G}_\gamma) + P(\overline{G} \notin \mathcal{G}_\gamma)P(h(\overline{G}) \neq f^*(\overline{G})|\overline{G} \notin \mathcal{G}_\gamma). \tag{9}$$

Furthermore, following (Devroye et al., 2013, Section 2.4), we have:

$$\epsilon^* = \frac{1}{2} - \frac{1}{2}\mathbb{E}(|2P_{Y|\overline{G}}(1|\overline{g}) - 1|) \Rightarrow \mathbb{E}(|2P_{Y|\overline{G}}(1|\overline{g}) - 1|) = 1 - 2\epsilon^*.$$

Consequently, by the Markov inequality, we have:

$$P(1 - |2P_{Y|\overline{G}}(1|\overline{g}) - 1| \geq \gamma) \leq \frac{2\epsilon^*}{\gamma}, \quad \gamma \in [0, 1]$$

So, $P(\overline{G} \in \mathcal{G}_\gamma) \leq \frac{2\epsilon^*}{\gamma}, \gamma \in [0, 1]$. Using equation 9, we get:

$$P(h(\overline{G}) \neq f^*(\overline{G})) \leq \frac{2\epsilon^*}{\gamma} + P(h(\overline{G}) \neq f^*(\overline{G}), \overline{G} \notin \mathcal{G}_\gamma), \tag{10}$$

On the other hand, using (Devroye et al., 2013, Theorem 2.2),

$$P(h(\overline{G}) \neq Y) = \epsilon^* + 2\sum_{\overline{g}} P_{\overline{G}}(\overline{g})|P_{Y|\overline{G}}(1|\overline{g}) - \frac{1}{2}|\mathbb{1}_{f(\overline{g}) \neq f^*(\overline{g})}$$

$$\geq \epsilon^* + 2\sum_{\overline{g} \notin \mathcal{G}_\gamma} P_{\overline{G}}(\overline{g})|P_{Y|\overline{G}}(1|\overline{g}) - \frac{1}{2}|\mathbb{1}_{f(\overline{g}) \neq f^*(\overline{g})}$$

$$\geq \epsilon^* + 2(1 - \gamma)\sum_{\overline{g} \notin \mathcal{G}_\gamma} P_{\overline{G}}(\overline{g})\mathbb{1}_{f(\overline{g}) \neq f^*(\overline{g})},$$

where in the last inequality, we have used the definition of $\mathcal{G}_\gamma$. Note that $\sum_{\overline{g} \notin \mathcal{G}_\gamma} P_{\overline{G}}(\overline{g})\mathbb{1}_{f(\overline{g}) \neq f^*(\overline{g})} = P(\overline{G} \notin \mathcal{G}_\gamma)P(f(\overline{G} \neq f^*(\overline{G})|\overline{G} \notin \mathcal{G}_\gamma)$. Consequently,

$$P(h(\overline{G}) \neq Y) \geq \epsilon^* + 2(1 - \gamma)P(\overline{G} \notin \mathcal{G}_\gamma)P(f(\overline{G} \neq f^*(\overline{G})|\overline{G} \notin \mathcal{G}_\gamma)$$

$$\Rightarrow P(\overline{G} \notin \mathcal{G}_\gamma)P(f(\overline{G} \neq f^*(\overline{G})|\overline{G} \notin \mathcal{G}_\gamma) \leq \frac{P(h(\overline{G}) \neq Y) - \epsilon^*}{2(1 - \gamma)} = \frac{\delta}{2(1 - \gamma)}.$$

Hence, from equation 10, we have:

$$P(h(\overline{G}) \neq f^*(\overline{G})) \leq \frac{2\epsilon^*}{\gamma} + \frac{\delta}{2(1 - \gamma)}.$$

Similarly,

$$P(f(\overline{G}) \neq f^*(\overline{G})) \leq \frac{2\epsilon^*}{\gamma} + \frac{\epsilon - \epsilon^*}{2(1 - \gamma)}.$$

From equation 8, we get:

$$P(h(\overline{G}) \neq f(\overline{G})) \leq \frac{4\epsilon^*}{\gamma} + \frac{\delta + \epsilon - \epsilon^*}{2(1 - \gamma)}.$$

Minimizing the right-hand-side over $\gamma \in [0, 1]$, we get $\gamma^* = \frac{2\left(4\epsilon^* + \sqrt{2}\sqrt{\epsilon^*(\delta + \epsilon - \epsilon^*)}\right)}{9\epsilon^* - \delta - \epsilon}$ given that $9\epsilon^* > e_{max}(\epsilon^*, \kappa) + \delta$. Hence,

$$P(h(\overline{G}) \neq f(\overline{G})) \leq \frac{\sqrt{2}\sqrt{\epsilon^*\xi}\left(64\epsilon^{*2} - 16\epsilon^*\xi + \xi^2\right)}{2\left(-8\epsilon^*\xi + 8\sqrt{2}\epsilon^*\sqrt{\epsilon^*\xi} + \sqrt{2}\xi\sqrt{\epsilon^*\xi}\right)}$$

$$= \frac{\left(64\epsilon^{*2} - 16\epsilon^*\xi + \xi^2\right)}{2\left(-4\sqrt{2}\sqrt{\epsilon^*\xi} + 8\epsilon^* + \xi\right)}$$

$$= \frac{(8\epsilon^* - \xi)^2}{2\left(2\sqrt{2}\epsilon^* - \sqrt{\xi}\right)^2}$$

$$= \frac{\left(2\sqrt{2}\epsilon^* + \sqrt{\xi}\right)^2}{2} \triangleq \tau,$$

where we have defined $\xi \triangleq \delta + \epsilon - \epsilon^*$. Using the data processing inequality and Fano's inequality, we get:

$$I(\widehat{Y}; \overline{G}|\mathbb{1}_\Psi(\overline{G})) \leq I(h(\overline{G}); \overline{G}|\mathbb{1}_\Psi(\overline{G})) \leq h_b(\tau) + \tau \log|\mathcal{Y}|.$$

This proof is completed by noting that $\epsilon \leq e_{max}(\epsilon^*, \kappa)$.

$\square$

## A.3 PROOF OF PROPOSITION 2

*Proof.* We need to show that the condition in equation 3 is satisfied for any pair of explanations in $\mathcal{S}$. We consider two cases:

**Case 1:** $g_{exp} \subseteq \Psi(\overline{g})$ for all $\overline{g}$ for which $g_{exp} \subseteq \overline{g}$, that is, whenever the motif $g_{exp}$ is in $\overline{G}$, then the explanation function $\Psi(\overline{G})$ outputs the motif along with potentially other irrelevant edges and vertices. In this case, it is straightforward to verify that $\widehat{P}(y) = 1$, $\widehat{P}^+(y) = 1 - \mathbb{1}(y = 1)$, and $\widehat{P}^-(y) = 1$ for all $y \in \{0, 1\}$. So, $Fid_+ = P(Y = 1)$, $Fid_- = 0$, and $Fid_\Delta = P(Y = 1)$. Furthermore, $I(\widehat{Y}; \overline{G}|\mathbb{1}_\Psi(\overline{G})) = 0$.

**Case 2:** If the exists $\overline{g}$ such that $g_{exp} \subseteq \overline{g}$ but $g_{exp} \not\subseteq \Psi(\overline{g})$, then for any $\overline{g}$ such that $g_{exp} \subseteq \overline{g}$ but $g_{exp} \not\subseteq \Psi(\overline{g})$, we have $\widehat{P}(1) = 1$, $\widehat{P}^+(y) = \mathbb{1}(g_{exp} \cap \Psi(\overline{g}) = \phi)$, and $\widehat{P}^-(y) = 0$. Otherwise, if $g_{exp} \not\subseteq \overline{g}$ we have $y = 0$ and $\widehat{P}(1) = 1$, $\widehat{P}^+(y) = 1$, and $\widehat{P}^-(y) = 1$. So, $Fid_+ = P(Y = 1) - P(Y = 1, g_{exp} \cap \Psi(\overline{G}) = \phi)$, $Fid_- = 0$, and $Fid_\Delta = P(Y = 1) - P(Y = 1, g_{exp} \cap \Psi(\overline{G}) = \phi)$. Furthermore,

$$I(\widehat{Y}; \overline{G}|\mathbb{1}_\Psi(\overline{G})) = H(\widehat{Y}|\mathbb{1}_\Psi(\overline{G})),$$

where we have used the fact that this is a deterministic classification task. Note that since the graph edges are pairwise independent, we have:

$$H(\widehat{Y}|\mathbb{1}_\Psi(\overline{G})) = P(g_{exp} \cap \Psi(\overline{g}) = \phi)H(\widehat{Y}) +$$
$$P(g_{exp} \cap \Psi(\overline{G}) \neq \phi)H(\widehat{Y}|\mathbb{1}_\Psi(\overline{G}), g_{exp} \cap \Psi(\overline{G}) \neq \phi).$$

The latter is an increasing function of $P(g_{exp} \cap \Psi(\overline{g}) = \phi)$.

It can be observed that $Fid_\Delta$ in a decreasing and $I(\widehat{Y}; \overline{G}|\mathbb{1}_\Psi(\overline{G}))$ an increasing function of $P(g_{exp} \cap \Psi(\overline{g}) = \phi)$. Consequently, the condition in equation 3 is satisfied. This completes the proof. $\square$

## A.4 PROOF OF PROPOSITION 3

*Proof.* We have:

$$\epsilon^* - \epsilon \leq \mathbb{E}(\widehat{P}(Y)) \leq \epsilon^*,$$

where the left-hand-side follows from equation 7 since:

$$\mathbb{E}(\widehat{P}(Y)) \geq P(\mathcal{G})\mathbb{E}(\widehat{P}(Y)|\mathcal{G}) = P(\mathcal{G})P(Y = f^*(\overline{G})|\mathcal{G}),$$

along with:

$$\epsilon^* \leq P(\mathcal{G})P(Y = f^*(\overline{G})|\mathcal{G}) + P(\mathcal{G}^c)$$
$$= (1 - \epsilon)P(Y = f^*(\overline{G})|\mathcal{G}) + \epsilon \Rightarrow P(Y = f^*(\overline{G})|\mathcal{G}) \geq \frac{\epsilon^* - \epsilon}{1 - \epsilon}.$$

Furthermore,

$$\mathbb{E}(\widehat{P}^{\alpha_1, +}(Y)) \leq P(\mathcal{G})\mathbb{E}(\widehat{P}^{\alpha_1, +}(Y)|\overline{G} \in \mathcal{G}) + P(\mathcal{G}^c)$$
$$\overset{(a)}{\leq} \frac{P(\mathcal{A})}{1 - \epsilon}\mathbb{E}(\widehat{P}^{\alpha_1, +}(Y)|\overline{G} \in \mathcal{G}, \mathcal{A}) + P(\mathcal{A}^c) + \epsilon$$
$$\overset{(b)}{\leq} \frac{P(\mathcal{A})}{1 - \epsilon}P(\mathcal{B}')\mathbb{E}(\widehat{P}^{\alpha_1, +}(Y)|\overline{G} \in \mathcal{G}, \mathcal{A}, \mathcal{B}) + P(\mathcal{A}^c) + 1 - P(\mathcal{B}') + \epsilon$$
$$\overset{(c)}{\leq} \frac{P(\mathcal{A})}{1 - \epsilon}\mathbb{E}(\widehat{P}^{\alpha_1, +}(Y)|\overline{G} \in \mathcal{G}, \mathcal{A}, \mathcal{B}) + P(\mathcal{A}^c) + 1 - P(\mathcal{B}') + \epsilon$$
$$\overset{(d)}{\leq} \frac{P(\mathcal{A})}{1 - \epsilon}((1 - p + p\max_y P_Y(y)) + 1 - (\frac{1}{k+1})^\delta) + P(\mathcal{A}^c) + 1 - P(\mathcal{B}') + \epsilon,$$

where $\mathcal{A}$ is the event that $\exists! y : \overline{g}_y \subseteq \overline{G}$, $\mathcal{B}$ is the event that the sampling operation removes more than $k$ edges, and $\mathcal{B}'$ is the event that in $n^2$ consecutive realizations of a Bernoulli variable with parameter $\alpha_1$, we get more than $k$ ones. In (a) we have used the fact that $P(\mathcal{G}, \mathcal{A}) \leq P(\mathcal{A})$. In (b) we have used the fact that $P(\mathcal{B}|\mathcal{A}, \mathcal{G}) \leq P(\mathcal{B}')$. In (c), we have used $P(\mathcal{B}') \leq 1$. In (d), we have used the

assumption in equation 7, and the fact that the explanation function does not output the motif with probability $(1-p)$, in which case it is not subtracted in the definition of $\widehat{P}^{\alpha_1,+}$, and it outputs the motif with probability $p$ in which case the motif is not present in $\overline{G} - E_{\alpha_1}(\Psi(\overline{G}))$ since $k < s_1$. Additionally,

$$\mathbb{E}(\widehat{P}^{\alpha_1,-}(Y)) \geq P(\mathcal{A})(1 - 2^{\frac{k^2}{2n^2}})(\frac{1}{k+1})^\delta p,$$

where we have used the fact that if $\mathcal{A}$ occurs, and the explanation outputs the motif, which happens with probability $p$, then the optimal classifier $f^*$ outputs the correct label, and if $\mathcal{B}'$ does not occur, then $f_\delta(\cdot)$ agrees with $f^*(\cdot)$ with probability $(\frac{1}{k+1})^\delta$, and we have bounded $1 - P(\mathcal{B}') \geq (1 - 2^{\frac{k^2}{2n^2}})$ by Hoeffding's inequality. Consequently,

$$Fid_{\alpha_1,+} \geq \epsilon^* - \epsilon - \frac{P(\mathcal{A})}{1-\epsilon}((1 - p + p\max_y P_Y(y)) + 1 - (\frac{1}{k+1})^\delta)$$
$$- P(\mathcal{A}^c) - 1 - P(\mathcal{B}') - \epsilon$$

$$Fid_{\alpha_2,-} \leq \epsilon^* - P(\mathcal{A})(1 - 2^{\frac{k^2}{2n^2}})(\frac{1}{k+1})^\delta p$$

$$Fid_{\alpha_1,\alpha_2,\Delta} \geq -\epsilon - P(\mathcal{A}^c) - 1 - P(\mathcal{B}') - \frac{P(\mathcal{A})}{1-\epsilon}((1 - p + p\max_y P_Y(y)) +$$
$$1 - (\frac{1}{k+1})^\delta) + P(\mathcal{A})(1 - 2^{\frac{k^2}{2n^2}})(\frac{1}{k+1})^\delta p.$$

Note that the lower-bound on $Fid_{\alpha_1,\alpha_2,\Delta}$ is monotonically increasing in $p$. To prove that $Fid_{\alpha_1,\alpha_2,\Delta}$ is well-behaved as $k, n \to \infty$ and $\delta, \epsilon \to 0$, it suffices to show that equation 3 holds. Note that:

$$I(Y;\overline{G}|\mathbb{1}_{\Psi(\overline{G})}) = \sum_{g_{exp}} P_{\Psi(\overline{G})}(g_{exp}) \sum_{y,\overline{g}} P_{Y,\overline{G}}(y,\overline{g}|g_{exp} \subseteq \overline{G}) \log \frac{P_{Y,\overline{G}}(y,\overline{g}|g_{exp} \subseteq \overline{G})}{P_Y(y|g_{exp} \subseteq \overline{G})P_{\overline{G}}(\overline{g}|g_{exp} \subseteq \overline{G})}$$
$$= P_{\Psi(\overline{G})}(\phi)I(Y;\overline{G}) + \sum_{y\in\mathcal{Y}} P_{\Psi(\overline{G})}(g_y)I(Y;\overline{G}|g_y \subseteq \overline{G}). \tag{11}$$

From the proposition statement, we have:

$$P_{\Psi(\overline{G})}(\phi) = P(\nexists! y : g_y \subseteq \overline{G}) + (1-p)P(\exists! y : g_y \subseteq \overline{G}) \tag{12}$$
$$P_{\Psi(\overline{G})}(g_y) = pP(\exists! y' : g_{y'} \subseteq \overline{G}, y' = y), \quad \forall y \in \mathcal{Y}. \tag{13}$$

As a result, from equations 11-13, we have:

$$I(Y;\overline{G}|\mathbb{1}_{\Psi(\overline{G})}) = P(\nexists! y : g_y \subseteq \overline{G})I(Y;\overline{G}) +$$
$$\sum_{y\in\mathcal{Y}} P(\exists! y' : g_{y'} \subseteq \overline{G}, y' = y)((1-p)I(Y;\overline{G}) + pI(Y;\overline{G}|g_y \subseteq \overline{G})),$$

where we have used the fact that $P(\exists! y : g_y \subseteq \overline{G}) = \sum_{y'\in\mathcal{Y}} P(\exists! y' : g_{y'} \subseteq \overline{G}, y' = y)$. We note that $I(Y;\overline{G}) \to H(Y)$ and $I(Y;\overline{G}|g_y \subseteq \overline{G}) \to 0$ as $\epsilon^* \to 0$. So, there exists an error threshold $\epsilon^*$ such that for all $\epsilon^* \leq \epsilon_{th}$ and $y \in \mathcal{Y}$, the term $(1-p)I(Y;\overline{G}) + pI(Y;\overline{G}|g_y \subseteq \overline{G})$ is decreasing as a function of $p$. Since the lower-bound on $Fid_{\alpha_1,\alpha_2,\Delta}$ is monotonically increasing in $p$, and $I(Y;\overline{G}|\mathbb{1}_{\Psi(\overline{G})})$ is monotonically decreasing in $p$, we conclude that equation 3 holds and $Fid_{\alpha_1,\alpha_2,\Delta}$ is well-behaved. $\square$

## B  EXAMPLES AND ANALYSIS

**An Analytical Example with Distribution Shift for the $Fid_\Delta$ Measure:**  In this example, we provide an asymptotically deterministic classification scenario where for an errorless classifier $f(\cdot)$, there exists an explanation function $\Psi_1$ which is asymptotically optimal in the sense of Definition 3, but has a worse fidelity score under $Fid_\Delta$ compared to a classifier $\Psi_2$ which generates completely random outputs. We conclude that $Fid_\Delta$ is not well-behaved for this scenario. To elaborate, let us

consider a classification scenario where $\overline{G}$ has $n \in \mathbb{N}$ vertices, and it can be decomposed as the union of two subgraphs $\overline{G}_0$ and $\overline{G}_{exp}$. Further assume that $\overline{G}_0$ is Erdös-Renyi with edge probability $p \in (0,1)$ and has $n \in \mathbb{N}$ vertices. The graph $\overline{G}_{exp}$ is equal to $C_n$ with probability $q \in (0,1)$, and it is equal to an empty graph with probability $1-q$, where $C_n$ is the $n$-cycle $v_1 \to v_2 \to \cdots \to v_n \to v_1$. We call the graph $\overline{G}$ atypical if it has less than than $\frac{n^2 p}{4}$ edges. Note that the expected number of edges in $\overline{G}_0$ is $\frac{n(n-1)}{2}p$, and by the law of large numbers, as $n \to \infty$, the graph $\overline{G}$ has more than $\frac{n^2 p}{4}$ edges with probability one. Let us assume that $Y = 1$ if $\overline{G}$ is typical and $C_n \subseteq \overline{G}$, and $Y = 0$, otherwise. Let us consider the asymptotically optimal classifier $f(\overline{G})$ which outputs $\widehat{Y} = 1$ if $C_n \subseteq \overline{G}$ and $\overline{G}$ is not atypical, and outputs $\widehat{Y} = 0$, otherwise. Consider the explanation function

$$\Psi_1(\overline{G}) = \begin{cases} C_n & \text{if } C_n \subseteq \overline{G} \\ \phi & \text{Otherwise.} \end{cases}$$

Clearly, we have $I(\widehat{Y}; \overline{G} | \mathbb{1}_{\Psi(\overline{G})}) \to 0$ as $n \to \infty$, and hence $\Psi(\cdot)$ is an (optimal) $(n, 0)$-explanation function as $n \to \infty$. Note that, $Fid_+(f, \Psi_1) \to 1 - P(Y = 0)$, $Fid_-(f, \Psi_1) \to 1 - P(Y = 0)$, and $Fid_\Delta(f, \Psi_1) \to 0$ as $n \to \infty$. On the other hand, let $\Psi_2(\overline{G})$ be the explanation function which outputs a randomly and uniformly chosen subset of more than $\frac{n^2 p}{4}$ edges from $\overline{G}$. Then $Fid_\Delta(f, \Psi_2) = P(Y = 1) - P(Y = 0) = 2q - 1$ which is greater than 0 if $q > 1/2$. So, the trivial explanation $\Psi_2$ receives a higher score under $Fid_\Delta$ compared to the optimal explanation $\Psi_1$.

## C  EVALUATION ALGORITHMS

In this section, we provide the pseudo-code for computing the proposed $Fid_{\alpha_1,+}$ and $Fid_{\alpha_2,-}$ in Alg. 1 and Alg. 2, respectively. Suppose that we have a set of input graphs, $\{\overline{G}_i\}_{i=1}^{\mathcal{T}}$. For each graph $\overline{G}_i$, the explanation subgraph to be evaluated is denoted by $\Psi(\overline{G}_i)$. The model to be explained is denoted by $f(\cdot)$. We have two hyper-parameters, $M$ and $\alpha_1$ in computing $Fid_{\alpha_1,+}$. $M$ is the number of samples and $\alpha_1$, introduced in equation 5, is the ratio of edges sampled from explanation subgraph. For $Fid_{\alpha_2,-}$, we have another hyper-parameter $\alpha_2$ instead, which indicates the ratio of edges sampled from non-explanation subgraph.

---

**Algorithm 1** Computating $Fid_{\alpha_1,+}$

---

1: **Input:** A set of input graphs and their subgraphs $\{(\overline{G}_i, \Psi(\overline{G}_i)\}_{i=1}^{\mathcal{T}}$, a GNN model $f(\cdot)$, hyperparameters $M$ and $\alpha_1$.
2: **Output:** $Fid_{\alpha_1,+}$ of $\{\Psi(\overline{G}_i)\}_{i=1}^{\mathcal{T}}$.
3: **for** each pair $(\overline{G}_i, \Psi(\overline{G}_i))$ **do**
4:    **for** $m$ from 1 to $M$ **do**
5:       $E_{\alpha_1}(\Psi(\overline{G}_i))) \leftarrow$ sample $\alpha_1$ edges from $\Psi(\overline{G}_i)$
6:       $\overline{G}_{i,m} \leftarrow \overline{G}_i - E_{\alpha_1}(\Psi(\overline{G}_i)))$      # Compute the non-explanation subgraph
7:       $Fid_{\alpha_1+}[i,m] \leftarrow f(\overline{G}_i)_{y_i} - f(\overline{G}_{i,m})_{y_i}$
8:    **end for**
9:    $Fid_{\alpha_1+}[i] \leftarrow \frac{1}{M} \sum_{m=1}^{M} Fid_{\alpha_1,+}[i,m]$
10: **end for**
11: $Fid_{\alpha_1,+} \leftarrow \frac{1}{\mathcal{T}} \sum_{i=1}^{\mathcal{T}} Fid_{\alpha_1,+}[i]$
12: **Return** $Fid_{\alpha_1,+}$.

---

## D  EXTENSION BEYOND GRAPHS

Some efforts have been conducted to address the distribution shift problem in evaluating general XAI methods (Hooker et al., 2019). Most of them focus on designing more robust perturbation-based explanatory methods (Hase et al., 2021; Hsieh et al., 2021; Qiu et al., 2021; Jethani et al., 2023). For example, ROAR retrains the model with a modified dataset to ensure the generalization of the classification model (Hooker et al., 2019). Robustness measurements include adversarial perturbations into metric (Hsieh et al., 2021). However, these methods are designed for grid data and it is non-trivial to adapt them to graphs. Moreover, these methods require access to the internal parameters of classification models and are not applicable to black-box models.

---

**Algorithm 2** Computating $Fid_{\alpha_2,-}$

---

1: **Input:** A set of input graphs and their subgraphs $\{(\overline{G}_i, \Psi(\overline{G}_i)\}_{i=1}^{\mathcal{T}}$, a GNN model $f(\cdot)$, hyperparameters $M$
   and $\alpha_2$.
2: **Output:** $Fid_{\alpha_2,-}$ of $\{\Psi(\overline{G}_i)\}_{i=1}^{\mathcal{T}}$.
3: **for** each pair $(\overline{G}_i, \Psi(\overline{G}_i))$ **do**
4:     **for** $m$ from 1 to $M$ **do**
5:         $\overline{G}_i^c \leftarrow \overline{G}_i - \Psi(\overline{G}_i)$             # Compute the non-explanation subgraph
6:         $E_{\alpha_2}(\overline{G}_i^c) \leftarrow$ sample $\alpha_2$ edges from $\overline{G}_i^c$
7:         $Fid_{\alpha_2,-}[i,m] \leftarrow f(\overline{G}_i)_{y_i} - f(E_{\alpha_2}(\overline{G}_i^c) + \Psi(\overline{G}_i))_{y_i}$
8:     **end for**
9:     $Fid_{\alpha_2,-}[i] \leftarrow \frac{1}{M} \sum_{m=1}^{M} Fid_{\alpha_2,-}[i,m]$
10: **end for**
11: $Fid_{\alpha_2,-} \leftarrow \frac{1}{\mathcal{T}} \sum_{i=1}^{\mathcal{T}} Fid_{\alpha_2,-}[i]$
12: **Return** $Fid_{\alpha_2,-}$.

---

In this study, we focus on the graph domain. The notion of explainability considered in this work is specific to sub-graph explanations. Recall that a graph $g_{exp}$ is a subgraph of $g$ if its vertices can be 'aligned' with a subset of vertices in $g$ such that all edges between the two sets of vertices match. The subgraph explainability is key in the definition of mutual information $I(Y; \overline{G}|\mathbb{1}_{\Psi(\overline{G})})$, which is then used to define explainability in Definitions 2 and 3. As a basic foundation, unfortunately, the definition of mutual information is non-trivial to be extended to general domains. However, the underlying ideas considered in this paper are general and may be applicable to non-graphical tasks, which is an interesting avenue for future research.

## E    DETAILED EXPERIMENTAL SETUP

We use a Linux machine with 8 NVIDIA A100 GPUs as the hardware platform, each with 40GB of memory. The software environment is with CUDA 11.3, Python 3.7.16, and Pytorch 1.12.1.

### E.1    DATASETS

We adopt two node classification datasets and two graph classification datasets with ground-truth motifs. Specifically, the Tree-Cycles dataset includes an 8-depth balanced binary tree as the base graph. 80 cycle motifs are then randomly attached to nodes from the base graph. (2) Tree-Grid is created in a similar way, except that the cycle motifs are replaced with 3-by-3 grids. (3) The BA-2motifs dataset consists of 1000 graphs. Half of them are obtained by attaching a 'house' motif to a 20-node Barabási–Albert random graph. Other graphs are with 5-node cycle motifs. Graphs are divided into 2 classes based on the type of attached motifs. For these datasets, we use a 10-dimensional all 1 vector as node attributes (Ying et al., 2019). We also use a real-life dataset, MUTAG, for graph classification, which consists of 4,337 molecular graphs. These graphs are labeled based on their mutagenic effects (Ying et al., 2019). As discussed in (Luo et al., 2020), chemical groups $NH_2$ or $NO_2$ are used as ground truth motifs.

### E.2    CLASSIFICATION MODEL ARCHITECTURES

The GCN model architectures and hyperparameters of layers are the same as the previous work (Luo et al., 2020). Specifically, for the node classification, a two-stack GCN-Relu-BatchNorm block followed by a GCN-Relu block is used for embedding. A following linear layer is used for classification. For the graph classification, a three-stack GCN-Relu-BatchNorm block is used for compute node embedding. Global max and mean pooling are used to read out the graph representations. A linear layer is then used for classification. Similarly, we create GIN models by replacing the GCN layer with a Linear-Relu-Linear-Relu GIN layer. All the variables are initialized with the default setting in Pytorch. The models are trained with the Adam optimizer with an initial learning rate of $1.0 \times 10^{-3}$. For each dataset, we follow existing works (Luo et al., 2020; Ying et al., 2019) to split train/validation/test with $8 : 1 : 1$ for all datasets. Each model is trained for 1000 epochs.

## F EXTRA EXPERIMENTAL STUDIES

### F.1 EFFECTS OF $\alpha_1$ AND $\alpha_2$.

As shown in our theoretical analysis, $\alpha_1$ is the rate of removing edges from explanation subgraphs in $Fid_{\alpha_1,+}$ and $\alpha_2$ is the rate of retaining edges from non-explanation subgraphs in $Fid_{\alpha_2,-}$. To empirically verify the effects of these two parameters, we use the GCN model and vary these two hyper-parameters in the range $[0.1, 0.3, 0.5, 0.7, 0.9]$. Results of Spearman correlation scores are shown in Figure 1. We observe that when $\alpha_1 = 0.1$ and $\alpha_2 = 0.9$, the proposed $Fid_{\alpha_1,+}$ and $Fid_{\alpha_2,-}$ are strongly aligned with the gold standard edit distance. As $\alpha_1$ increases, the number of removing edges from explanation subgraphs increases, leading to more a severe distribution shifting problem. Thus, the Spearman correlation coefficient between $Fid_{\alpha_1,+}$ and edit distance increases. Specifically, in the Tree-Circles dataset, the correlation even becomes positive when $\alpha_1 > 0.2$. The similar phenomena can be observed in $Fid_{\alpha_2,-}$. As $\alpha_2$ decreases, a smaller number of edges will be added from non-explanation subgraphs, which leads to the distribution shift problem. As a result, the $Fid_{\alpha_2,-}$ cannot be reliably aligned with the gold standard metric.

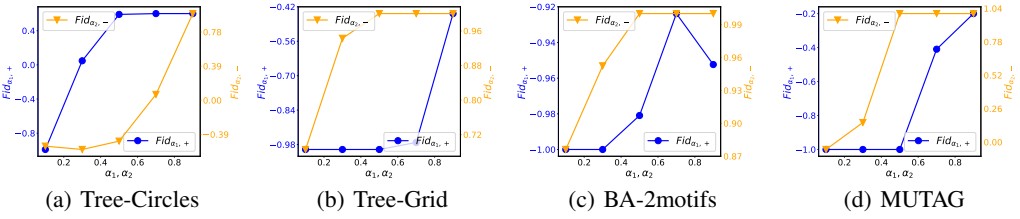

|     (a) Tree-Circles     |     (b) Tree-Grid     |     (c) BA-2motifs     |     (d) MUTAG     |

Figure 1: Parameter studies on the effects of $\alpha_1$ and $\alpha_2$.

### F.2 CLASSIFICATIONS PERFORMANCE OF GNNS

We evaluate the accuracy performance of GNN models on training, validation, and test sets. The results are shown in Table 2. Both GCN and GIN achieve good performances in these datasets, with most test accuracy scores above 0.9. Following routinely adopted settings (Ying et al., 2019; Luo et al., 2020; Yuan et al., 2021), we can safely assume that both models can correctly use the informative components(motifs) in the input graphs to make predictions.

Table 2: Accuracy performance of GNN models.

| Method | Accuracy | Node Classification | | Graph Classification | |
| --- | --- | --- | --- | --- | --- |
| | | Tree-Cycles | Tree-Grid | BA-2motifs | MUTAG |
| GCN | Training | 0.99 | 0.92 | 1.00 | 0.82 |
| | Validation | 1.00 | 0.94 | 1.00 | 0.82 |
| | Test | 0.99 | 0.94 | 0.99 | 0.81 |
| GIN | Training | 1.00 | 0.98 | 0.99 | 0.86 |
| | Validation | 1.00 | 0.99 | 1.00 | 0.84 |
| | Test | 0.99 | 0.97 | 1.00 | 0.82 |

### F.3 DISTRIBUTION ANALYSIS

In Section 3.2.1, we argued that the previously studied $Fid_{\Delta}$, $Fid_-$ and $Fid_+$ surrogate measures suffer from the OOD problem, since $Fid_-$ and $Fid_+$ rely on accurate classifier outputs for inputs $\Psi(\overline{G})$ and $\overline{G} - \Psi(\overline{G})$, respectively, and $Fid_{\Delta}$ depends on both. In this section, we empirically verify the aforementioned OOD problem. Specifically, we adopt an AutoEncoder (Kramer, 1991) to embed the adjacency matrix of a (sub-)graph to a 2-dimensional hidden representation. We choose the graph classification dataset, BA-2motifs, where all graphs have the same node size. The encoder network consists of two fully-connected layers, same as that of the decoder network. Cross Entropy is used as the reconstruction error to train the Autoencoder model. The original graphs, the ground-truth explanation subgraphs, and the non-explanation subgraphs are used for training.

The visualization results of these three types of (sub-)graphs are shown in Figure 2(a). We observe that both explanation subgraphs and non-explanation subgraphs have clear distribution shifts compared to the original graphs.

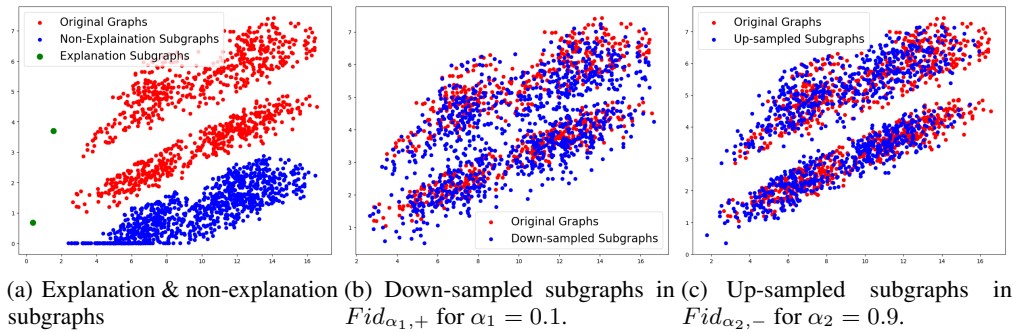

(a) Explanation & non-explanation subgraphs
(b) Down-sampled subgraphs in $Fid_{\alpha_1,+}$ for $\alpha_1 = 0.1$.
(c) Up-sampled subgraphs in $Fid_{\alpha_2,-}$ for $\alpha_2 = 0.9$.

Figure 2: Distribution analysis with visualization.

The proposed methods address the challenge by considering 'down/up-sampled subgraphs'. Specifically, in $Fid_{\alpha_1,+}$, given a graph $\overline{G}$ and an explanation subgraph $\Psi(\overline{G})$, as shown in Algorithm 1, we randomly remove $\alpha_1$ ratio of edges in $\Psi(\overline{G})$ from $\overline{G}$ and compare the prediction with the original one. For simplicity, we denote this as a 'down-sampled' subgraph. For $\alpha_1 = 0.1$, Figure 2(b) shows the visualization results of these 'down-sampled' subgraphs. We observe that these subgraphs are in-distributed. Similarly, in $Fid_{\alpha_2,-}$, we sample $\alpha_2$ ratio of edges from the non-explanation subgraph and add them to the explanation subgraph. We denote this graph as an 'up-sampled' subgraph. The visualization results with $\alpha_2 = 0.9$ are shown in Figure 2(c), which also verifies the in-distribution property.

To quantitatively evaluate the distribution shifting problem, we first normalize the representation vector and use KL-divergence to measure the distance between up/down-sampled subgraphs and the original graphs. The KL divergence between down-sampled subgraphs and original graphs with respect to $\alpha_1$ is shown in Figure 3(a). When $\alpha_1 = 1.0$, the down-sampled subgraphs degenerate into non-explanation subgraphs, where the KL divergence is large. For a small value of $\alpha_1$, the KL divergence is small. In Figure 3(b), we show the KL divergence between up-sampled subgraphs and original graphs with respect to $\alpha_2$. When $\alpha_2 = 0.0$, the up-sampled subgraphs become explanation subgraphs, which are out-of-distributed, as evidenced by a large KL divergence value. On the other hand, with a large value of $\alpha_2$, for example 0.9, we are able to largely alleviate the OOD problem.

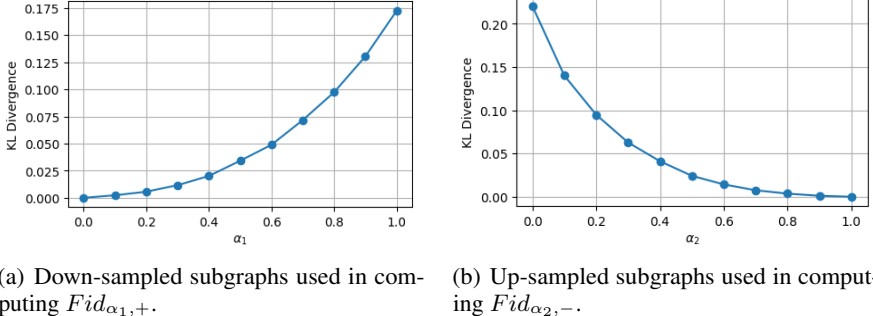

(a) Down-sampled subgraphs used in computing $Fid_{\alpha_1,+}$.
(b) Up-sampled subgraphs used in computing $Fid_{\alpha_2,-}$.

Figure 3: KL divergence between down/up-sampled subgraphs and original graphs.

### F.4 STABILITY ANALYSIS

In the proposed fidelity measurements, we sample $M$ times to compute $Fid_{\alpha_1,+}$ and $Fid_{\alpha_2,-}$, respectively. To verify the stability of our measurements. We change the value of $M$ in the range

$(10, 100)$ and keep $\alpha_1 = 0.1$ and $\alpha_2 = 0.9$. For each value of $M$, we evaluate the quality of ground-truth explanations 10 times and report the mean as well as the standard deviation in Fig. 4. Specifically, for the default setting, $M = 50$, we also report the comparison of running time between our methods and their counterparts in Table 3.

We have the following observations. First, as $M$ increases, the proposed metrics are more stable. Second, our measurements are quite robust as the mean scores are stable with $M$ ranging from 10 to 100. Thus, although our method is slower than the original fidelity measurements, in practice, a small number of samples are enough to get precise estimations.

Table 3: Running time performance on GT explanation

| Model | Time(s) | Node Classification | | Graph Classification | |
| --- | --- | --- | --- | --- | --- |
| | | Tree-Circles | Tree-Grid | BA-2motifs | MUTAG |
| GCN | Ori. | 1.06 | 3.77 | 2.98 | 56.91 |
| | Ours | 13.34 | 64.44 | 48.97 | 302.38 |
| GIN | Ori. | 0.63 | 2.71 | 2.12 | 50.99 |
| | Ours | 7.74 | 37.65 | 30.53 | 198.93 |

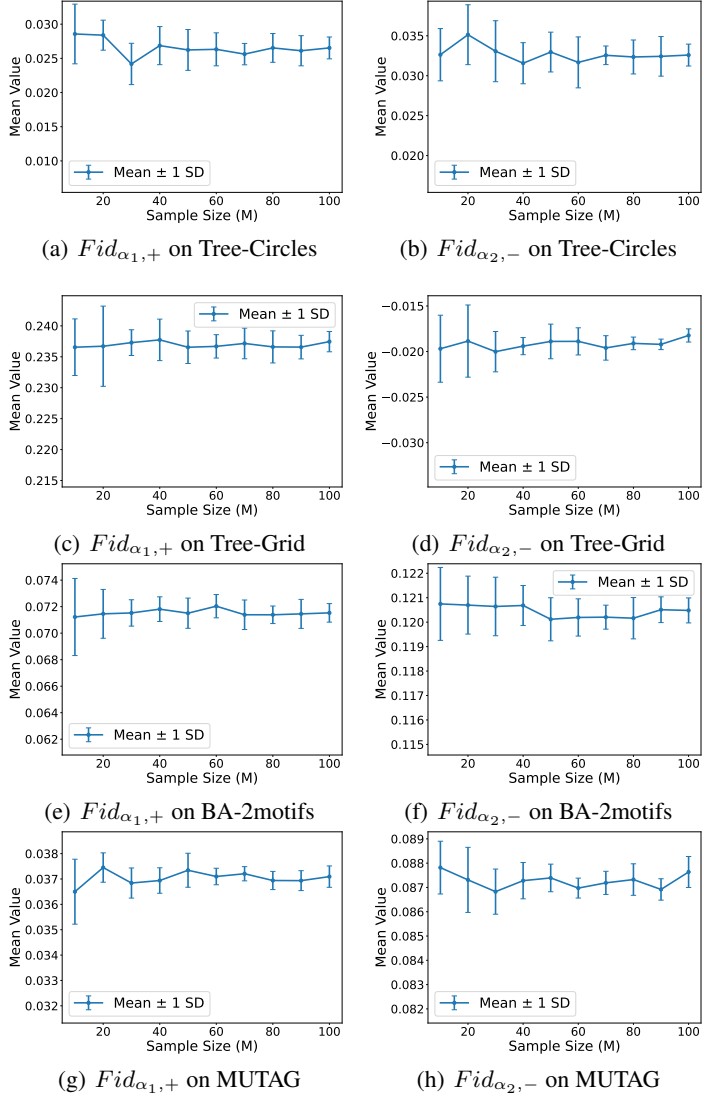

(a) $Fid_{\alpha_1,+}$ on Tree-Circles

(b) $Fid_{\alpha_2,-}$ on Tree-Circles

(c) $Fid_{\alpha_1,+}$ on Tree-Grid

(d) $Fid_{\alpha_2,-}$ on Tree-Grid

(e) $Fid_{\alpha_1,+}$ on BA-2motifs

(f) $Fid_{\alpha_2,-}$ on BA-2motifs

(g) $Fid_{\alpha_1,+}$ on MUTAG

(h) $Fid_{\alpha_2,-}$ on MUTAG

Figure 4: Parameter studies on $M$.

F.5  QUALITATIVE EVALUATION

To qualitatively show the quality of different fidelity metrics, in this part, we visualize the fidelity scores with heat maps. We adopt the GCN model and follow the experimental setting in Sec.5.1. Results are shown in Fig. 5, 6, 7, and 8. We can observe several phenomena. First, with the same $\beta_2$, a candidate explanation generated by a small $\beta_1$ indicates a smaller edit distance and a better explanation quality. Thus, for $Fid_+$, the values should decrease as $\alpha_1$ increases. However, as shown in Fig. 5(a), 6(a), 7(a), and 8(a), the original $Fid_+$ fails to keep the monotonicity. For example, when looking at the first column of Fig. 7(a), we observe that the $Fid_+$ scores first go up and then go down with $\beta_1$ increasing and it achieves the highest score when there are 50% edges removed from the ground-truth explanations. The result shows that $Fid_+$ fails to measure the correctness of explanations. On the other hand, the new proposed $Fid_{\alpha_1,+}$ is highly monotonic to gold standard edit distance. We have similar observations with $Fid_-$ and $Fid_\Delta$ on all these datasets.

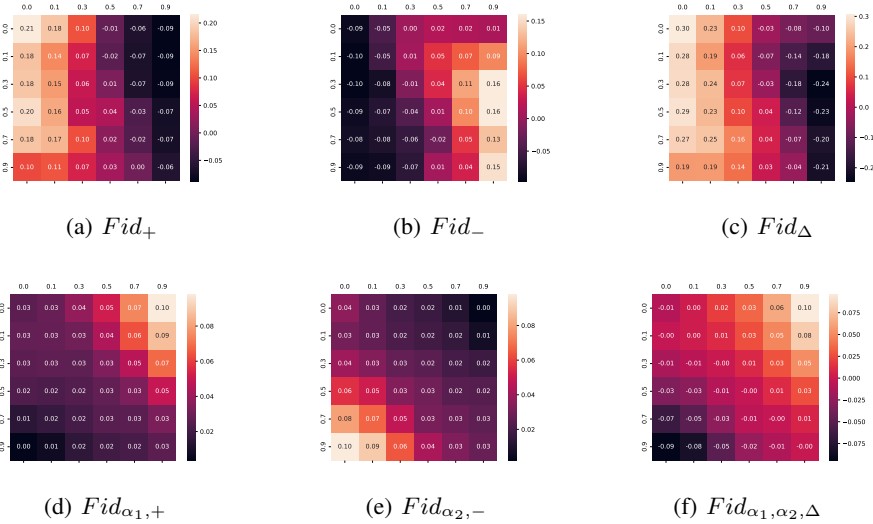

Figure 5: Fidelity scores with GCN on Tree-Circles. The X-axis value, $\beta_2$ is the ratio of added edges from the non-explanation subgraph to the candidate explanation. The Y-axis value, $\beta_1$ is the ratio of edges removed from ground truth in the candidate explanation.

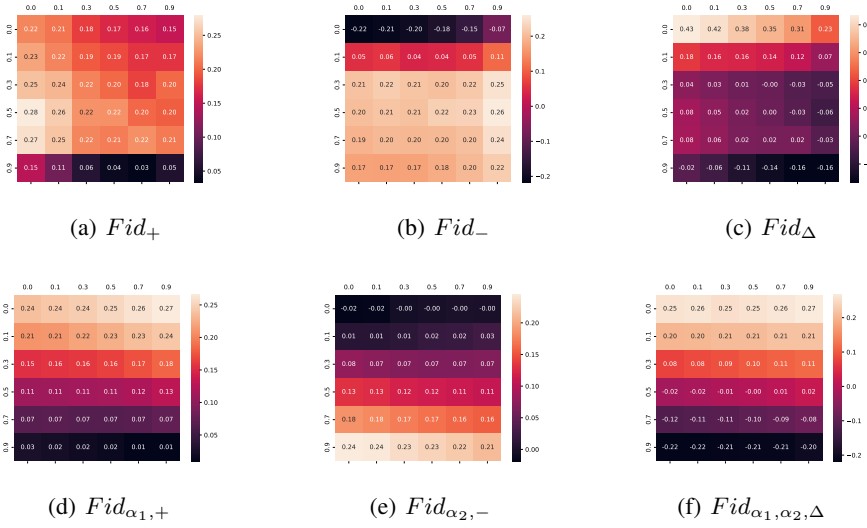

Figure 6: Fidelity scores with GCN on Tree-Grid.

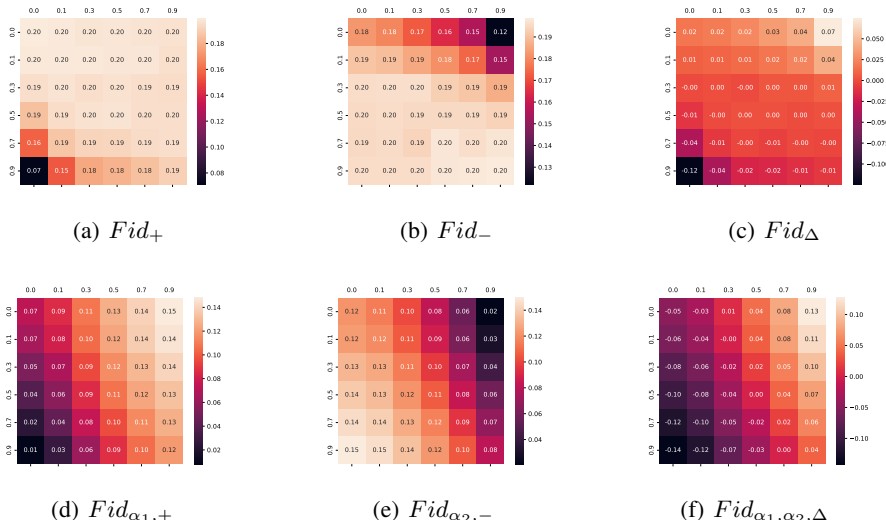

Figure 7: Fidelity scores with GCN on BA-2motifs.

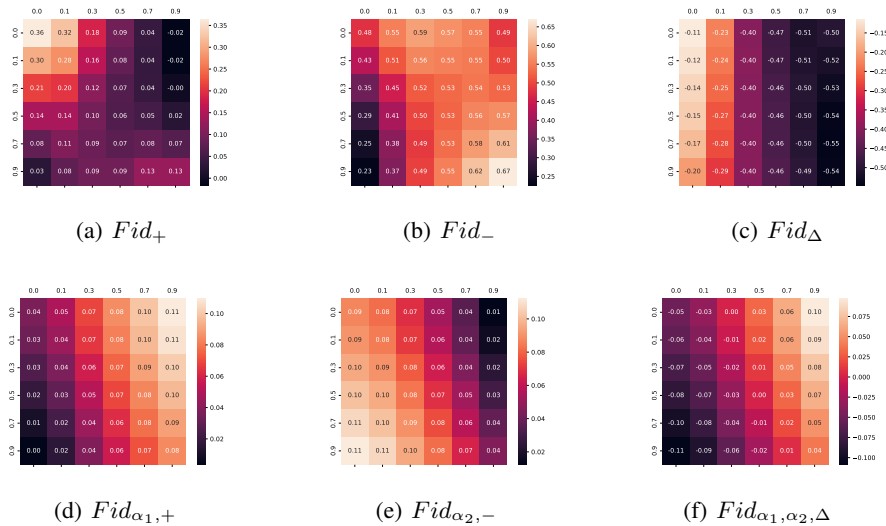

Figure 8: Fidelity scores with GCN on MUTAG.

## F.6 CASE STUDIES

In this section, we adopt case studies to show the effects of the OOD problem in original fidelity measurement and the robustness of the new proposed metrics. We use both graph classification datasets, BA-2motifs and MUTAG in this part. For each dataset, we choose a graph. We consider two candidate explanations: the gold standard one and an adversarial explanation. As shown in Table 4, the first row is the gold standard explanation and the second row is an adversarial explanation with an edit distance of 10, which doesn't contain the label information. The $Fid_+$ scores of these two explanations are close. Moreover, $Fid_-$ and $Fid_\Delta$ mistakenly suggest that the adversarial explanation is better. These results further verify that the original fidelity metrics are fragile. On the other hand, the proposed fidelity metrics are more robust. All $Fid_{\alpha_1,-}$, $Fid_{\alpha_2,-}$, and $Fid_{\alpha_1,\alpha_2,\Delta}$ successfully give the correct measurements. We have a similar conclusion in the case study of MUTAG in Table 5.

Table 4: A case study on BA-2motifs dataset. The bold font indicates the better explanation subgraph suggested by fidelity measurements.

| Explanation | Edit Dis. ↓ | $Fid_+$ ↑ | $Fid_-$ ↓ | $Fid_\Delta$ ↑ | $Fid_{\alpha_1,+}$ ↑ | $Fid_{\alpha_2,-}$ ↓ | $Fid_{\alpha_1,\alpha_2,\Delta}$ ↑ |
|---|---|---|---|---|---|---|---|
|  | 0 | -0.012 | -0.036 | 0.024 | **0.169** | **0.002** | **0.167** |
|  | 10 | **-0.010** | **-0.052** | **0.042** | 0.009 | 0.103 | -0.095 |

Table 5: A case study on MUTAG dataset. The bold font indicates the better explanation subgraph suggested by fidelity measurements.

| Explanation | Edit Dis. ↓ | $Fid_+$ ↑ | $Fid_-$ ↓ | $Fid_\Delta$ ↑ | $Fid_{\alpha_1,+}$ ↑ | $Fid_{\alpha_2,-}$ ↓ | $Fid_{\alpha_1,\alpha_2,\Delta}$ ↑ |
|---|---|---|---|---|---|---|---|
|  | 0 | 0.619 | 0.627 | -0.008 | **0.092** | **0.303** | **-0.211** |
|  | 10 | **0.697** | **-0.030** | **0.727** | 0.056 | 0.352 | -0.296 |

## F.7 ACCURACY BASED FIDELITY

As another variant, accuracy-based Fidelity measurement is also used to evaluate the performance of explanations (Yuan et al., 2022). Let use $\{(\bar{G}_i, y_i)\}_{i=1}^{\mathcal{T}}$ denote a set of $\mathcal{T}$ of graphs and their labels. $f(\cdot)$ is the GNN classifier to be explained. For the $i$-th graph $\bar{G}_i$, $\Psi(\bar{G}_i)$ denotes the explanation subgraph output by an explainer $\Psi(\cdot)$ acting on $\bar{G}_i$. The accuracy-based fidelities are defined as follows.

$$Fid_+^{(acc)} = \frac{1}{\mathcal{T}} \sum_{i=1}^{\mathcal{T}} \left| \mathbb{1}(\arg\max f(\bar{G}_i) = y_i) - \mathbb{1}(\arg\max f(\bar{G}_i - \Psi(\bar{G}_i)) = y_i) \right|$$

$$Fid_-^{(acc)} = \frac{1}{\mathcal{T}} \sum_{i=1}^{\mathcal{T}} \left| \mathbb{1}(\arg\max f(\bar{G}_i) = y_i) - \mathbb{1}(\arg\max f(\Psi(\bar{G}_i)) = y_i) \right| \tag{14}$$

$$Fid_\Delta^{(acc)} = Fid_+^{(acc)} - Fid_-^{(acc)}$$

In a similar way, we formulate our accuracy-based Fidelity measurements, $Fid_{\alpha_1,+}^{(acc)}$, $Fid_{\alpha_2,-}^{(acc)}$ and $Fid_{\alpha_1,\alpha_2,\Delta}^{(acc)}$. We adopt the same setting as Sec. F.5 and show the visualization results in Fig 9,10,11, and12. We observe consistent improvements achieved by our methods over their counterparts, indicating their effectiveness in the accuracy-based setting.

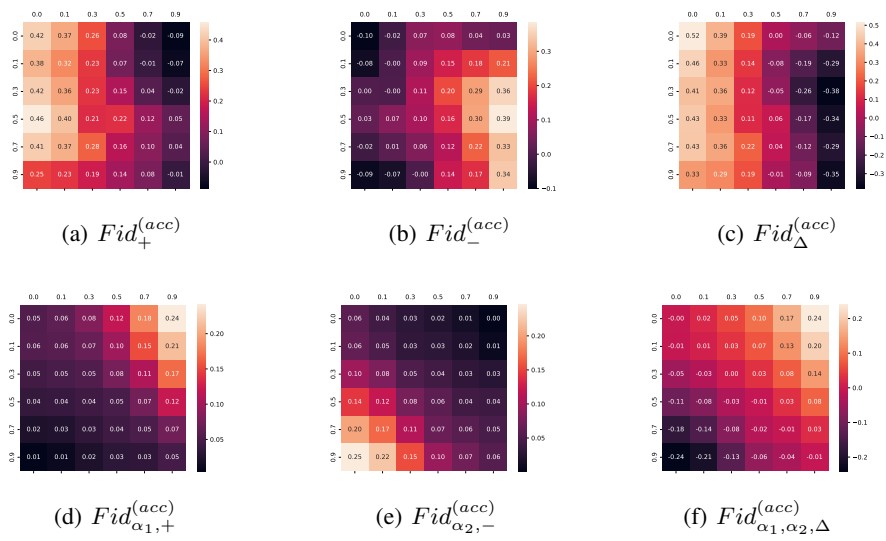

Figure 9: Accuracy-based Fidelity results of GCN on Tree-Circles.

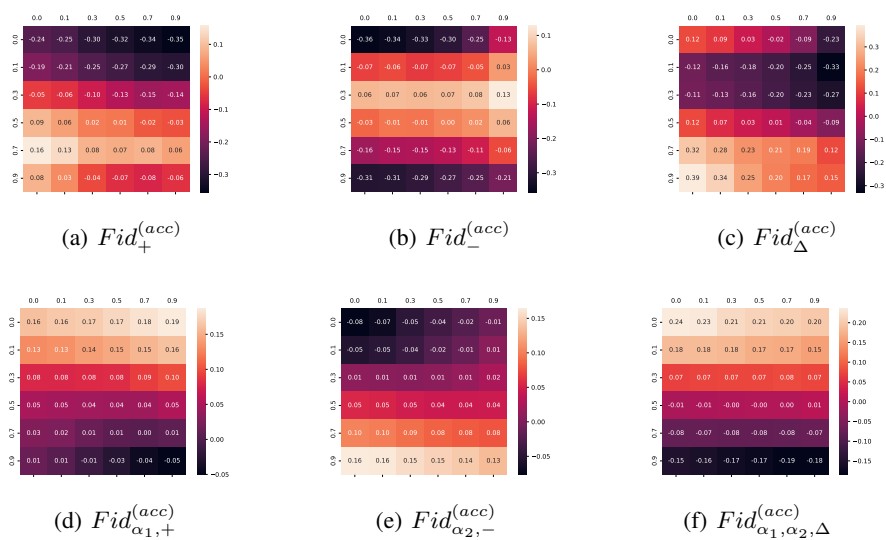

Figure 10: Accuracy-based Fidelity results of GCN on Tree-Grid.

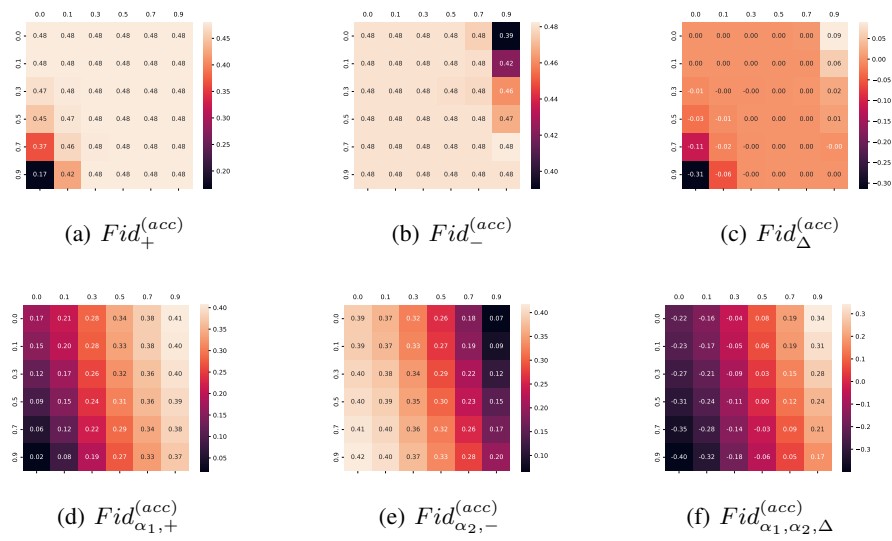

Figure 11: Accuracy-based Fidelity results of GCN on BA-2motifs.

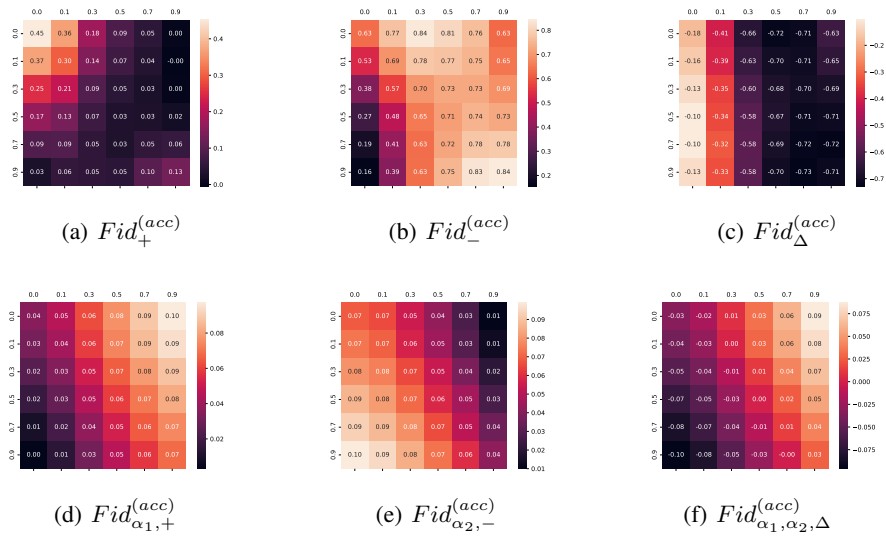

Figure 12: Accuracy-based Fidelity results of GCN on MUTAG.

