# OpenReview forum: "Towards Robust Fidelity for Evaluating Explainability of Graph Neural Networks"
_ICLR.cc/2024/Conference — ICLR 2024 poster_

### Official Review · Reviewer_aFFv · 2023-10-28

**Soundness:** 3 good
**Presentation:** 3 good
**Contribution:** 3 good
**Rating:** 8
**Confidence:** 4

**Summary:**

The paper tackles the crucial challenge of evaluating the fidelity of explanations provided by Graph Neural Networks (GNNs). Given the increasing importance of GNNs in sensitive sectors, understanding their decision-making process is paramount. Traditionally, fidelity metrics like $Fid_+$, $Fid_-$, and $Fid_\Delta$ have been utilized to measure the correctness of these explanations. The intuition behind these metrics is to assess the change in model predictions when certain subgraphs, deemed important for predictions, are masked out or removed. However, the authors spotlight potential issues with these metrics, chiefly the distribution shift when subgraphs are removed. An information-theoretic framework for GNN explainability is presented, highlighting the misalignment of prevalent surrogate fidelity metrics with their proposed fidelity measure. As a solution, a novel class of fidelity measures that are robust to distribution shifts is introduced. The paper validates these measures through empirical analysis on synthetic and real datasets, demonstrating that the new metrics align more closely with ground truth explanations.

**Strengths:**

Originality: The paper effectively spotlights an overlooked issue in the field of explainable graph learning, bringing to the fore the inherent shortcomings of widely accepted evaluation methodologies. The introduction of robust evaluation metrics grounded in information theory provides a fresh perspective on the problem.

Quality: The paper maintains high standards in its methodological approach. It's evident that thorough theorical analysis and experiments has been done.

Clarity: Overall, the paper is well organized that logically from identifying the problem, proposing a theoretical framework, and then introducing new evaluation metrics. It would be better if the authors can use a figure to clearly illusrate the differences between original fidelity and the proposed ones.

Significance: This research is timely and significant. Given the increasing number of papers on explainable GNNs. The routinely adopted metric is problematic and maybe misleading.  The paper's proposed fidelity metrics could play a important role in the literature.

**Weaknesses:**

1. The proposed metrics are not easy to understand until reading the algorithm in appendix.

2. Only small scale datasets are adopted to evaluate the proposed metrics.

3. It is unclear why we need to consider the rate of convergence in fidelity measurements.

**Questions:**

1. On Page 5, the authors claims that the convergence ratio of fidelity is 1/2. What does that mean in practice?  Does the proposed metric provide better convergence rate?

2. The paper focus on graph classification problem. But in the experiment, there are two node classification datasets, how can this method be applied for node classification task?

3. is OOD problem unique to graph? Or it also affects other domains, like images and natural languages.

4. Are there computational complexities or scalability concerns with the new metrics, especially when dealing with large real-world graphs?

---

> ### Author Response · Authors · 2023-11-19
> **Official response to reviewer aFFv (Part 1/2)**
>
> Dear Reviewer aFFv,
>
> Thank you for your valuable comments on our work. Below are our responses.
>
> > W1 The proposed metrics are not easy to understand until reading the algorithm in the appendix.
>
> Thanks for the comment. To address the reviewer's concern, we have split the method section into two subsections.  We also include more discussions in the related work part.
>
> The current organization of the paper is as follows:
> - Section 2: Provides brief formulation of the graph and node classification tasks.
> - Section 2.1: Provides an information theoretic quantification of explainability of tasks and classifiers (Definitions 2 and 3).
> - Section 3.1: Shows that the explainability of a classification task is equivalent to explainability of a `good' classifier for that task (Theorem 1). This is a fundamental question and the answer allows us to conclude that since the tasks considered in simulations of Section 5 are explainable, "good" classifiers for those tasks should also be explainable, hence they should yield high fidelity values if the measure used to evaluate is well-behaved.
> - Section 3.2: Argues that while the information theoretic measures of Section 2.1 provide a theoretical foundation, and allow us to answer the question posed in Section 3.1 as discussed above, they are not helpful in practice due to high sample complexity in their estimation e.g., [1]. This leads us to define surrogate fidelity measures which are empirically estimatable (Definitions 4 and 5).
> - Section 3.2.1: Considers a previously studied surrogate fidelity measure ($Fid_\Delta$ considered in GraphXAI) and shows that it is well-behaved when applied to deterministic classification tasks. However, we argue that it suffers from OOD issues and is not generally well-behaved for non-deterministic tasks.
> - Section 3.2.2: Introduces a new surrogate fidelity measure which resolves the OOD problem in the original GraphXAI measure (Proposition 3).
> - Section 5: The newly introduced surrogate fidelity measure is applied to various synthetic and real-world datasets in Section 5, and it is shown that it improves upon the GraphXAI surrogate fidelity measures, as predicted by our theoretical derivations.
>
> [1] McAllester, David, and Karl Stratos. "Formal limitations on the measurement of mutual information." International Conference on Artificial Intelligence and Statistics. PMLR, 2020.
>
>
> > W2 Only small scale datasets are adopted to evaluate the proposed metrics.
>
> Thanks for the comment. We adopt the routinely adopted benchmark datasets in the literature, including two node classification and two graph classification datasets. Among these datasets, the MUTAG dataset consists of over 4000 graphs.
>
> > W3. It is unclear why we need to consider the rate of convergence in fidelity measurements.
>
> The rate of convergence indicates whether the fidelity measure can be accurately evaluated/estimated using a limited training set, with a faster rate of convergence indicating that the evaluation can be performed accurately on a smaller training set.
>
> To elaborate, note that the parametrization of explainability provided in Definitions 2 and 3 captures the intuitive notion of explainability. However, as observed in prior literature on GraphXAI, these mutual-information-based measures, whose evaluation requires estimating a mutual-information term, cannot be evaluated accurately using a limited training set (e.g., [1]). Consequently, prior works have considered the $Fid_{\Delta}$ measure as a measure of explainability as discussed in Section 3.2. The $Fid_{\Delta}$ measure, has rate of convergence $\frac{1}{2}$, and hence can be estimated using a limited, but large enough training set; however, it suffers from the OOD issue. We propose a new set of evaluation measures, which also have rate of convergence $\frac{1}{2}$ and can be evaluated using a limited training set, but which do not suffer from the OOD issue.
>
>
> > Q1. On Page 5, the authors claim that the convergence ratio of fidelity is 1/2. What does that mean in practice? Does the proposed metric provide better convergence rate?
>
> The significance of showing that the rate of convergence of the proposed measures is  $\frac{1}{2}$ is that this is the same rate of convergence achieved by the previously studied $Fid_{\Delta}$ measure. So, we can claim that while we resolve the OOD issue to some extent, we are not increasing the necessary size of the training set for evaluating the fidelity metrics by proposing the modified measures. To summarize, we show that we achieve the same rate of convergence as the previous fidelity measure while improving in the sense of OOD.

---

> ### Author Response · Authors · 2023-11-19
> **Official response to reviewer aFFv (Part 2/2)**
>
> > Q2 The paper focus on graph classification problem. But in the experiment, there are two node classification datasets, how can this method be applied for node classification task?
>
> As we discussed in Section 2, "In node classification tasks, each graph $G_i$ denotes a $K$-hop sub-graph centered around node $v_i$, with a GNN model $f$ trained to predict the label for node $v_i$ based on the node representation of $v_i$ learned from $G_i$." Intuitively, we transform the node classification problem into its $K$-hop sub-graph classification problem, which is a widely used technique in the literature, such as GNNExplainer and PGExplainer.
>
>
>
>
> > Q3 is OOD problem unique to graph? Or it also affects other domains, like images and natural languages.
>
>
> A comprehensive discussion on extension of th ideas discussed in the paper to non-graphical domians is provided in Appendix D in the revised manuscript. Additional explanation in response to the reviewer's comment is provided below.
>
> The underlying ideas considered in this paper, such as the ones related to the OOD issue, may be relevant to non-graphical tasks. However, the application of the specific techniques and arguments provided here to more general domains is non-trivial, and is an interesting avenue for future research. Particularly, Definitions 2 and 3, which form the foundation of our work, and provide a rigorous information theoretic definition of explainability in graphs, are based on the notion "sub-graph explanations". To extend the observations of the paper to non-graphical domains, one needs to provide analogous definitions which are not based on the notion of sub-graph explanations.
>
> To elaborate further and provide additional explanation, let us consider an image classification task, where the objective is to detect the presence or absence of an object, e.g., a ball, in the input image. Consider a generic explanation method used in such image classification scenarios, such as SHAP (SHapley Additive exPlanations) and LIME (Local Interpretable Model-agnostic Explanations) methods. Both methods can be used to generate a "heatmap" signifying the influence of each of the pixels in generating the classification output. The heatmap can be used to choose the most important "k" pixels in the image, similar to how the most important edges of a graph are chosen by graph explanation methods considered in this work. In this case, one can take the most important "k" pixels as the "sub-image" and use padding to turn it into a valid input to the classifier. Under this procedure, it should be expected that that the OOD issue would manifest. However, this definition of a "sub-image explanation" is fragile and would not be robust to affine transformations such as rotation, scaling, and shifts. So, to provide analogous analysis as those provided in our work, one needs to develop more robust notions of "sub-image explanations". This is an interesting avenue for future research and the authors will work on extending their results to non-graphical domains in the future.
>
>
>
>
> > Q4 Are there computational complexities or scalability concerns with the new metrics, especially when dealing with large real-world graphs?
>
> Thanks for the comment. Our method is efficient because we don't re-train the classification model.  Since we need sampling to calculate the proposed fidelity measurements, we admit that the proposed method is still slower the original fidelity computation, when applied to the same finite training set. However, as we analyzed in Figure 2,  our method can achieve consistent results with a small number of samples.
>
> Furthermore, we argue in Section 3.2 that the rate of convergence in computing our proposed metric is $\frac{1}{2}$, similar to that of the $Fid_{\Delta}$ metric studied in prior works. So, in the asymptotic regime (with vanishing estimation errors), empirical estimation of the new fidelity metric would require the training sets of similar size, and same computational complexity as that of  $Fid_{\Delta}$ .
>
>
>
> Should you have any further questions or require additional information, please feel free to contact us.

---

> ### Author Response · Authors · 2023-11-22
> **Follow up**
>
> Dear Reviewer aFFv,
>
> Thank you for your review and constructive comments on our paper. We have provided corresponding responses, which we believe have covered your concerns. As the discussion period is close to its end, we would appreciate your feedback on whether or not our responses sufficiently resolve your concerns. Please let us know if you still have any concerns that we can address.
>
> Best regards,

---

### Official Review · Reviewer_oeQs · 2023-10-30

**Soundness:** 4 excellent
**Presentation:** 3 good
**Contribution:** 4 excellent
**Rating:** 8
**Confidence:** 4

**Summary:**

This paper points out a crucial problem in evaluating the explainability of Graph Neural Networks (GNNs) that the subgraph is distributed differently from the training graphs. Recognizing the limitations of conventional fidelity metrics ($Fid_+$, $Fid_-$, and $Fid_\Delta$) in capturing genuine model explainability due to the potential distribution shifts, the authors introduce an information-theoretic definition of explainability, carefully design and propose a straight forward evaluation methods by adopting sampling. They use 4 datasets and two tasks in the experimental part. They show that the proposed method is more consistent to ground truth ones.

**Strengths:**

1. The research problem is significant in the graph learning domain. They show that the existing evaluation method is heavily affected by distribution shifting problem, which is overlooked by existing methods. The proposed evaluation method has the potential to set new standards in the evaluation of GNN explainability.
2. The paper provides strong theoretical analysis and empirical verification.
3. The paper provides a simple and effective method with solid theoretical foundation.

**Weaknesses:**

I  have a minor concern that the proposed evaluation method is not deterministic and more time consuming comparing to the original fidelity measurement.

Minor typos.

1. In the introduction, "nodes presenting atoms" should likely be "nodes representing atoms".
2. In the introduction, "subgraph of the input which satisfies two conditions" shoule be "subgraph of the input that satisfies two conditions"
3. In Section 3, "low error probability" should be "a low error probability", "with error close"-> "with an error close", " has Bayes error rate" -> " has a Bayes error rate"

**Questions:**

1. Why the proposed method fails with GIN on Tree-grid?
2. How to select $alpha_1$ and $alpha_2$?

---

> ### Author Response · Authors · 2023-11-19
> **Official response to reviewer oeQs**
>
> Dear Reviewer oeQs,
>
> Thank you for your valuable comments on our work. Below are our responses.
>
> > Q1 Why the proposed method fails with GIN on Tree-grid?
>
> As we discussed on Page 9, we observed that all fidelity measurements fail to evaluate the GIN classifier on the Tree-Grid dataset. The potential reason is that GIN was designed for graph classification and its generalization performance on node classification is limited in the Tree-Grid dataset.
>
> >2 How to select $\alpha_1$ and $\alpha_2$?
>
> $\alpha_1$ is the parameter in $Fid_+$, $\alpha_2$ is the parameter in $Fid_-$. Both values are in the range of $[0,1]$. As we discussed on Page 6, when $\alpha_1=1$ and $\alpha_2=0$, the proposed fidelity measurements degenerate to the original $Fid_+$ and $Fid_-$. Our theoretical analysis suggests that we choose a small value for $\alpha_1$ and a large value for $\alpha_2$. We adopt $\alpha_1=0.1$ and $\alpha_2=0.9$ in our experiments, which achieves consistent performance.  We also conduct parameter sensitivity in Figure 1, which further verifies our selection of $\alpha_1$ and $\alpha_2$.
>
> We also take the opportunity to fix typos.  Thank you for recognizing the significance of our paper.

---

> > ### Comment · Reviewer_oeQs · 2023-11-22
> >
> > Dear authors, sorry for the late reply, and thanks for your answer.  I appreciate the clarifications.

---

> ### Author Response · Authors · 2023-11-22
> **Follow up**
>
> Dear Reviewer oeQs,
>
> Thank you for your review and constructive comments on our paper. We have provided corresponding responses, which we believe have covered your concerns. As the discussion period is close to its end, we would appreciate your feedback on whether or not our responses sufficiently resolve your concerns. Please let us know if you still have any concerns that we can address.
>
> Best regards,

---

### Official Review · Reviewer_gPjw · 2023-10-30

**Soundness:** 3 good
**Presentation:** 3 good
**Contribution:** 3 good
**Rating:** 5
**Confidence:** 4

**Summary:**

The present work studies existing fidelity metrics to quantify performance of GNN explanation methods. Through a theoretical analysis, the authors find that existing fidelity metrics are not well-behaved for a wide set of scenarios without special properties due to the OOD nature of explanation subgraphs. To address this, the authors proposed modified fidelity metrics where explanation subgraphs are transformed to approximate the underlying data distribution and hence obtain more accurate approximations of predictions from trained GNN models, which, the authors argue, provide well-behaved proxy fidelity metrics.

**Strengths:**

-	Addressing a fundamental aspect of GNN explainability, i.e., how well current metrics evaluate GNN explanation methods. This is a crucial topic.
-	Theoretical and empirical analysis to show that the widely accepted fidelity metrics are not well behaved for a wide range of scenarios without special properties.
-	Innovative proposal to transform sampled explanation subgraphs closer to the distribution of graphs in the training set for a well-behaved measure of fidelity.

**Weaknesses:**

The authors mention that current fidelity metrics are not well behaved because optimized models learn how to approximate the (graph) data distribution when making predictions and explanation subgraphs are rarely part of the dataset on their own, resulting in (potentially) poor estimates of predictions for explanation subgraphs. To address this, the authors design fidelity metrics which add/remove edges such that the explanation subgraphs approximate the training data distribution during evaluation of fidelity. I have three questions regarding this:
1.  What are the authors’ thoughts on how the nature of explanations should be? I think intuition says that explanation subgraphs should indeed be OOD w.r.t. the data distribution, after all, we’re looking for *sub*graphs that stand out (can be predictive on their own). Do authors generally think that the field should be moving to finding IID explanation subgraphs (if this is possible)?
2.  It appears that the OOD aspect is addressed in the proposed fidelity metrics. However, it seems to me that the sampling when computing fidelity metrics means that these metrics are not actually measuring fidelity of the actual explanation subgraph (but down/up-sampled versions of it). Why/how can we say that these fidelity metrics actually correspond to the explanation subgraphs themselves when these metrics measure fidelity *not* exactly for the explanation subgraphs?
3. A different direction could be to accept that predictions for OOD subgraphs are not as accurate and compute fidelity metrics and compare explanations for subgraphs with the same sparsity level, as is done in https://proceedings.neurips.cc/paper/2021/file/2c8c3a57383c63caef6724343eb62257-Paper.pdf What are the authors’ thoughts on this?

The authors have done a great job at studying how fidelity metrics are not well behaved for a wide range of scenarios, which is a crucial question in GraphXAI. I, however, have some questions regarding their solution to the OOD issue of explanation subgraphs, specifically, my question 2. If this (and other concerns) are addressed, I am open to updating my score.

**Questions:**

Could you elaborate on why proving that the proposed fidelity metrics are monotonically increasing in p means that they are monotonically increasing with the mutual information?

---

> ### Author Response · Authors · 2023-11-19
> **Official response to reviewer gPjw (Part 1/3)**
>
> Dear Reviewer gPjw,
>
> Thank you for your valuable comments on our work. Below are our responses.
>
> > W1. What are the authors’ thoughts on how the nature of explanations should be? I think intuition says that explanation subgraphs should indeed be OOD w.r.t. the data distribution, after all, we’re looking for subgraphs that stand out (can be predictive on their own). Do authors generally think that the field should be moving to finding IID explanation subgraphs (if this is possible)?
>
> We thank the reviewer for the interesting discussion regarding the nature of explanations. In the literature of Graph explainability, intuitively, an explanation is defined as a (small) subgraph that dominates the prediction. Building on this intuitive notion, in Definitions 2 and 3 in the manuscript, we have parametrized explainability using the following two quantities: (i) the explanation subgraph  $g\_{exp}(\mathcal{V}\_{exp}, \mathcal{E}\_{exp})$ is an (almost) sufficient statistic of $\overline{G}$ with respect to the true label $Y$, i.e., $I(Y; \overline{G}| {1}_{\Psi(\overline{G})})\leq \kappa$ for a small $\kappa>0$, and (ii) the explanation has a "small" number of edges on average $\mathbb{E}\_G(|\mathcal{E}\_{exp}|)\leq s$
> for a small $s\in \mathbb{N}$. As the reviewer has correctly pointed out, in most applications of interest, the second assumption renders the explanation OOD with respect to the actual data distribution, as the actual graph $\overline{G}$ is usually large, and has more than $s$ edges.
> In our view, condition (ii) is an essential component in defining explainability, as considered in the prior literature. The reviewer's suggestion, of moving towards a new notion of explainability is an interesting avenue for future research. To our understanding "IID explanations" in this context refer to explanations that are not out of distribution with respect to the original input distribution, e.g., graphs of similar size as the input.  In this case, one can view the explanation task as detecting or generating prototypes [1], which generates an in-distributed graph or selects a representative input as the explanation. We agree that finding such in-distributed explanation subgraphs is also important and promising, especially in model-level (class-level) explanations.  However, it should be noted that such a definition of explanation deviates from the current standard notion and would be an interesting avenue for future research. The authors would be very interested in exploring other notions of "IID explanations" that the reviewer may have had in consideration.
>
> [1] Shin, Yong-Min, Sun-Woo Kim, and Won-Yong Shin. "PAGE: Prototype-Based Model-Level Explanations for Graph Neural Networks." arXiv preprint arXiv:2210.17159 (2022).
>
>
> > W2. It appears that the OOD aspect is addressed in the proposed fidelity metrics. However, it seems to me that the sampling when computing fidelity metrics means that these metrics are not actually measuring the fidelity of the actual explanation subgraph (but down/up-sampled versions of it). Why/how can we say that these fidelity metrics actually correspond to the explanation subgraphs themselves when these metrics measure fidelity not exactly for the explanation subgraphs?
>
> We thank the reviewer for the insightful question. We provide an intuitive analysis here. Our response consists of two parts, firstly, we provide an argument to show that the fidelity measures introduced in this work in Equations (5) and (6) are in agreement with the notion of explainability considered in the literature under some generic assumptions, secondly, we provide an argument as to why by looking at the randomly and uniformly down/up-sampled input graphs, we can evaluate the suitability of the explanation subgraphs.
>
> (To be continued)

---

> ### Author Response · Authors · 2023-11-19
> **Official response to reviewer gPjw (Part 2/3)**
>
> We base our response on the following three facts and intuitive assumptions:
>
> Fact 1: For a good classifier (with low error probability), the output must be correct for most "typical" inputs, where by "typical" we loosely mean inputs that are not extremely improbable.
>
> Fact 2: For most "typical" inputs, slight perturbations of those inputs are also "typical", e.g., if the graph is randomly and uniformly down/up-sampled by a small fraction of edges, it remains typical with high probability.
>
> Fact 3: For a good explanation, perturbing non-explanation edges in the input graph does not alter the true label, whereas perturbing the explanation edges would potentially alter the true label.
>
> Fact (3) holds since as mentioned in the response to Comment 1, the notion of explainability, studied in the GraphXAI literature, considers a subgraph as an explanation if it dominates the prediction.
>
> Let us assume Facts (1)-(3), and let $g_{exp}$ be a good explanation for $\overline{G}$. Furthermore,  let $\overline{G}'$ be  a slight perturbation of $\overline{G}$ containing $g_{exp}$ in the up-sampled case ($Fid_{\alpha_2,-}$), and containing a sampled version of $g_{exp}$ in the down-sampled case ($Fid_{\alpha_1,+}$).
>
> Let us first focus on the up-sampled case ($Fid_{\alpha_2,-}$). In this case, the true label of $\overline{G}'$ is the same as that of $\overline{G}$ from Fact (3) , additionally, from Fact (2), we conclude that $\overline{G}'$ is typical if $\overline{G}$ is typical, and from Fact (1), since $\overline{G}'$ is typical, a good model must output the correct label for $\overline{G}'$ with high probability which is the same label as that of $\overline{G}$. Hence yielding a small $Fid_{\alpha_2,-}$.
>
> On the other hand, in the  down-sampled case ($Fid_{\alpha_1,+}$),   the true label of $\overline{G}'$ is potentially different from that of $\overline{G}$ from Fact (3), additionally,  from Fact (2), $\overline{G}'$ is typical if $\overline{G}$ is typical, and from Fact (1), since $\overline{G}'$ is typical, a good model must output the correct label for $\overline{G}'$ with high probability which is potentially different from that of $\overline{G}$. Hence yielding a large $Fid_{\alpha_1,+}$.
>
> We conclude that for good explanations, $Fid_{\alpha_1,\alpha_2,\Delta}= Fid_{\alpha_1,+}-Fid_{\alpha_2,-}$ must be large, as long as $1-\alpha_2$, (intuitively, the perturbation ratio) is small and $\alpha_1$ is small.
>
> On the other hand, if $Fid_{\alpha_1,\alpha_2,\Delta}$ is not large for some small $1-\alpha_2$ and $\alpha_1$ and a given explanation subgraph, that means that for a non-negligible fraction of typical inputs, slight perturbations of edges not in the subgraph lead to different output predictions (so that $Fid_{\alpha_2,-}$ is large), and/or slight perturbations of edges in the subgraph does not alter the output (so that $Fid_{\alpha_1,+}$ is small). From Facts (1)-(3) one can conclude that the subgraph is  not a good explanation subgraph, given that the prediction model is a good model.
>
> Consequently, we conclude that given a good model, the explanation is good if and only if $Fid_{\alpha_1,\alpha_2,\Delta}$ is large. This confirms the well-behavedness of the measures introduced in the paper.
>
> Now we return to the question posed by the reviewer, which is, why is it that although the fidelity measures defined in Equations (5) and (6) only consider down/up-sampled input graphs, they are able to evaluate the suitability of the actual explanation graph as argued above.
>
> The reason lies in the application of Facts (2) and (3) in our argument above. Fact (2) is based on the assumption that the down/up-sampling operations (perturbations) are chosen randomly and uniformly. Note that in most applications of interest, it would be possible to choose a specific slight (adversarial) perturbation of a typical input to produce a non-typical input. This has been a topic of interest in the study of adversarial ML. However, by forcing a random and uniform perturbation, we ensure that the perturbed input is typical with high probability. Fact (3) is based on the assumption that the perturbation leaves the explanation graph unchanged in the case of $Fid_{\alpha_2,-}$, so that the prediction output is the same as that of the original graph, whereas it changes the explanation graph in the case of $Fid_{\alpha_1,+}$, so that the predication output is potentially different from that of the original graph.

---

> ### Author Response · Authors · 2023-11-19
> **Official response to reviewer gPjw (Part 3/3)**
>
> > W3. A different direction could be to accept that predictions for OOD subgraphs are not as accurate and compute fidelity metrics and compare explanations for subgraphs with the same sparsity level, as is done in https://proceedings.neurips.cc/paper/2021/file/2c8c3a57383c63caef6724343eb62257-Paper.pdf What are the authors’ thoughts on this?
>
>
> We thank the reviewer for bringing this important paper to our attention. We have added this reference when introducing $Fid_+$ in the introduction. The fidelity metric in the mentioned paper is similar to the $Fid_{\alpha_1,+}$ measure in Equation (5), and the previously studied $Fid_{+}$ measure discussed in Section 3.2.1 of the revised manuscript. As we discuss in the introduction, ``$Fid_+$ is defined as the difference in accuracy (or predicted probability) between the original predictions and the new predictions after masking out important input subgraphs''.
>
> Let use denote the original graph by $\overline{G}$, the explanation subgraph by $\Psi(\overline{G})$, and the GNN model by $f(\cdot)$. Then, $Fid_+$ measures the difference between $f(\overline{G})$ and $f(\overline{G}-\Psi(\overline{G}))$. We note that this computation is also affected by the OOD of $\overline{G}-\Psi(\overline{G})$. That is, the prediction of $f(G-\Psi(G))$ is not accurate if $\overline{G}-\Psi(\overline{G})$ is OOD. To elaborate, by restricting attention to $Fid_+$, the OOD issue of  $\Psi(\overline{G})$ does not manifest since it is not used as input to the classifier in computing $Fid_{+}$. However, it should be noted that $\overline{G}-\Psi(\overline{G})$ can still have OOD issues. The fidelity measures introduced in this paper resolve both of these OOD issues by introducing the sampling variables $\alpha_1,\alpha_2$.
>
>
> > Q1. Could you elaborate on why proving that the proposed fidelity metrics are monotonically increasing in p means that they are monotonically increasing with the mutual information?
>
> We thank the reviewer for encouraging us to provide the complete proof for the well-behavedness of $Fid_{\alpha_1,\alpha_2,\Delta}$ in Proposition 3. We have provided complete proof in Appendix A.4 in the revised manuscript. In the appendix we show that the mutual information is decreasing in p, hence Equation (3) holds, and $Fid_{\alpha_1,\alpha_2,\Delta}$ is well-behaved. It should be noted that Equation (3) defines well-behavedness as the property that the fidelity metrics be inversely related with the mutual information, e.g., in our example, the fidelity metric is increasing in $p$ and the mutual information is decreasing in $p$, so the fidelity metric is well-behaved.
>
> Note that our high-level objective in proving Proposition 3 is to show that the proposed fidelity metrics are well-behaved for a class of non-deterministic tasks as opposed to the previously studied $Fid_{\Delta}$ metric. We have added an explicit necessary condition in the statement of Proposition 3, which essentially states that the optimal Bayes classifier must not be too large (otherwise the task itself is not explainable as discussed in Section 3.1 and Theorem 1). The fact that the error can be non-zero proves our claim that  the proposed fidelity metrics are well-behaved in a class of non-deterministic tasks.
>
> Should you have any further questions or require additional information, please feel free to contact us.

---

> ### Author Response · Authors · 2023-11-21
> **A kind reminer**
>
> Dear reviewer gPjw,
>
> we submitted the reply a few days ago. Now the deadline for public comment is approaching. We are keen to ensure that our revisions and responses align with your expectations and address all your concerns effectively. Please feel free to let us know if you have other questions.

---

> ### Author Response · Authors · 2023-11-22
> **Final Discussion Time**
>
> Dear reviewer gPjw
>
> The rebuttal phase ends today and I haven't received feedback from you. We believe that we have addressed your previous concern about the rationality behind the usage of up/downsampled subgraphs. We have also added two more experiments to further support our claims. We would really appreciate that if you could check our response and updated paper.
>
> Looking forward to hearing back from you,
>
> Best Regards,
> Authors

---

> > ### Comment · Reviewer_gPjw · 2023-12-02
> > **Thanks for your responses**
> >
> > I apologize for the late reply which is not of much use after the rebuttal period has ended, I had multiple major commitments that took my time in the past couple of weeks.
> > Thank you for your thoughtful answers.
> >
> > Thanks for your response to W1. Regarding your response to W2, I follow the logic, but I see a key aspect that might require a different answer. Your answer covers the scenario of super/sub-sampling input graphs, and their effects on explanation subgraphs. In equations (5) and (6), what is being super/sub-sampled is the explanation subgraph itself, not the input, and I’m not sure how the argument you presented transfers to this case. Therefore, your comment doesn’t directly answer my question - again, sorry for the late reply, this is something that we maybe could’ve discussed further.
> >
> > Thank you for your responses to W3 and Q1, and the additional changes

---

### Official Review · Reviewer_dokf · 2023-11-01

**Soundness:** 2 fair
**Presentation:** 1 poor
**Contribution:** 2 fair
**Rating:** 3
**Confidence:** 4

**Summary:**

This paper analyzes the inherent limitations of prevailing fidelity metrics and proposes a robust class of fidelity measures.
The contributions are mainly about the relevant theoretical analysis

**Strengths:**

- Sufficient theoretical discussions regarding prevailing fidelity metrics are introduced in the paper

**Weaknesses:**

- The paper is poorly structured. The theoretical analysis took up too much space, and many of the derivations can be moved into the appendix. In the meantime, the discussion for the motivation of Fidelity measures is too limited.  Experimental results are also too limited.
- I hope the paper can better justify why the proposed fidelity metrics are ideal, and especially discuss their differences and relations with respect to the existing explainable GNN methods. Such discussions are lacking
- It is not apparent that why technical theoretical results are special for graphs. Many definitions and discussions seem to relate to general machine-learning problems. Can they be applied to data beyond graphs? If so, why the paper restrict the scope to graphs?

**Questions:**

- Can the proposed fidelity measures be applied to data beyond graphs? If so, why the paper restrict the scope to graphs?

---

> ### Author Response · Authors · 2023-11-19
> **Official response to reviewer dokf(Part 1/4)**
>
> Dear reviewer dokf,
> thank you for taking the time to review our work and providing feedback. In the following, we aim to address your questions and concerns.
>
> > W1. The paper is poorly structured. The theoretical analysis took up too much space, and many of the derivations can be moved into the appendix. In the meantime, the discussion for the motivation of Fidelity measures is too limited. Experimental results are also too limited.
>
> We thank the reviewer for the comment. We have made the following modifications to improve the readability and the overall organization of the paper:
>
>   - Added a descriptive title to each of the Proposition and Theorem statements in the paper, including Theorem 1, Proposition 1, Proposition 2, and Proposition 3.
>   - Separated Section 3 into several subsections, and modified the subsection titles for Sections 2 and 3 to clearly reflect the contents of each subsection.
>   - Add more discussion for the motivation of Fidelity measures in the introduction (para 2 and 3).
>   - Add more discussion of differences and relationships to the existing explainable GNN methods in
>     related work
>
> Please note that we have not included any theoretical derivations in the body of the original paper, and apart from the necessary definitions, problem statements, and proposition and theorem statements, all of the arguments and proofs are provided in the respective appendices.
>
> Through this paper, we have two main claims:
>
> 1) Existing fidelity measures are not well-behaved in a wide set of scenarios due to the OOD nature of subgraphs.
> 2) Our proposed fidelity measures, which are a natural result of our theoretical analysis and derivations, are more robust to OOD and are aligned well with gold-standard metrics.
>
> To demonstrate these claims, the current organization of the paper is:
> - Section 2: Provides brief formulation of graph and node classification tasks.
> - Section 2.1: Provides an information theoretic quantification of the explainability of tasks and classifiers (Definitions 2 and 3).
> - Section 3.1: Shows that the explainability of a classification task is equivalent to explainability of a "good" classifier for that task (Theorem 1). This is a fundamental question and the answer allows us to conclude that since the tasks considered in simulations of Section 5 are explainable, "good" classifiers for those tasks should also be explainable, hence they should yield good fidelity results if the measure used to evaluate is well-behaved.
> - Section 3.2: Argues that while the information theoretic measures of Section 2.1 provide a theoretical foundation, and allow us to answer the question posed in Section 3.1 as discussed above, they are not helpful in practice due to high sample complexity in their estimation e.g., [1]. This leads us to define surrogate fidelity measures which are empirically estimatable (Definitions 4 and 5).
> - Section 3.2.1: Considers a family of previously studied surrogate fidelity measures ($Fid_+, Fid_-, Fid_\Delta$) and shows that they are well-behaved when applied to deterministic classification tasks. However, we argue that they suffer from OOD issues and are not generally well-behaved for non-deterministic tasks.
> - Section 3.2.2: Introduces a new surrogate fidelity measure that resolves the OOD problem in the original measures (Proposition 3).
> - Section 4: introduce related works on explainable GNNs and discuss the relationship and differences to existing works.
> - Section 5: Empirically verify our claims on four benchmark datasets with two representative GNN models.
>
> To empirically verify our claims, extensive experiments are conducted in Section 5 and Appendix F:
>  - Quantitatively evaluate the proposed Fidelity metrics by comparing them to the gold standard on four benchmark datasets with two representative GNN models.
>  -  The proposed metrics have three hyper-parameters, namely $\alpha_1, \alpha_2$ and number of samples $M$.  We conduct parameter studies in Section 5.2 and F.3, respectively.  We also provide guidance on how to select reasonable hyper-parameters for real-life usage.
>  -  have added distribution analysis in Sec. F.2
>  -  Compare the running time of the proposed metrics with the original ones in Section F.3
>  -  Provide detailed fidelity scores with heat maps in Section F.4 to further verify the advantage of the proposed metrics
>  -  We have added case studies on two graph classification datasets in Section F.5 for a better understanding
>  -  Evaluate accuracy-based variants of $Fid_+$, $Fid_-$, and $Fid_\Delta$ in Section F.6
>
> [1] McAllester, David, and Karl Stratos. "Formal limitations on the measurement of mutual information." International Conference on Artificial Intelligence and Statistics. PMLR, 2020.
>
> We hope that the aforementioned modifications and explanation satisfactorily address the reviewer's concerns regarding the organization of the paper. We believe experiments are comprehensive to verify our main claims.

---

> ### Author Response · Authors · 2023-11-19
> **Official response to reviewer dokf(Part 2/4)**
>
> > W2 I hope the paper can better justify why the proposed fidelity metrics are ideal, and especially discuss their differences and relations with respect to the existing explainable GNN methods. Such discussions are lacking
>
> We thank the reviewer for pointing out the lack of a comprehensive discussion on the differences and relations with respect to the existing explainable GNN methods. We have included some discussion in the introduction and related work section (Section 4). Specifically, in the realm of explainable GNNs, both model design and evaluation are paramount. Most efforts have primarily been made to develop new network architectures and optimization objectives to achieve more accurate explanations. Orthogonal to these new explanation models, there have been some works on evaluating the achieved explanations, which is another fundamental research problem in the explainable GNN topic [1][2][3][4].  Most existing works, including regular papers [5] and platforms[1][2][3][4], utilize or include Fidelity as the main evaluation metric.  In this paper, we show that fidelity metrics are not well-behaved for a wide range of scenarios and propose a family of robust fidelity measures with information-theoretic guarantees. Thus, we believe that our work is significant and orthogonal to most existing explainable GNN methods. The proposed metrics can be included in existing evaluation platforms to reliably evaluate existing explainable GNN models.
>
>
> The second concern raised in the reviewer's comment is in regard to the justification of the proposed fidelity metrics, and the fact that in contrast with previous metrics, they are more robust to OOD issues. In response to this comment, please note that as discussed in Section 3.2.1, after the statement of Proposition 2, the previous fidelity metrics, ($Fid_{\Delta}, Fid_{+}, Fid_{-}$), rely on three main assumptions, namely, that a well-trained classifier provides accurate outputs when its input is i) the whole original graph, ii) only the explanation graph (for $Fid_{-}$ and $ Fid_{\Delta}$), and iii) the original graph with the explanation graph edges removed (for $Fid_{+}$ and $Fid_{\Delta}$). However, while Assumption i) is reasonable for well-trained classifiers, Assumptions ii) and iii) may not hold in general scenarios. In contrast, we prove that our proposed fidelity measures are more robust to OOD issues (Proposition 3). As discussed above, the reason for this robustness is the fact that these fidelity metrics do not rely on Assumptions ii) and iii), and only require that the classifier be accurate on slightly perturbed versions of `typically observed' graphs.  For a more detailed discussion of the latter statement, please refer to the first paragraph of Section 3.2.2. Furthermore,  we also empirically show that the proposed fidelity metrics are robust to the OOD problem and are aligned well with gold standard metrics in Section 5 where we generate candidate explanations for evaluation by edge sampling. By changing the sampling ratios, we can cover a wide range of candidate explanations to comprehensively evaluate the proposed metrics. Extensive experiments are also provided in Appendix F.
>
>
> [1] Liu, Meng, et al. "DIG: A turnkey library for diving into graph deep learning research." The Journal of Machine Learning Research 22.1 (2021): 10873-10881.
> [2] Amara, Kenza, et al. "Graphframex: Towards systematic evaluation of explainability methods for graph neural networks." arXiv preprint arXiv:2206.09677 (2022).
> [3] Kosan, Mert, et al. "GNNX-BENCH: Unravelling the Utility of Perturbation-based GNN Explainers through In-depth Benchmarking." arXiv preprint arXiv:2310.01794 (2023).
> [4] Agarwal, Chirag, et al. "Evaluating explainability for graph neural networks." Scientific Data 10.1 (2023): 144.
> [5] Yuan, Hao, et al. "Explainability in graph neural networks: A taxonomic survey." IEEE transactions on pattern analysis and machine intelligence 45.5 (2022): 5782-5799.

---

> ### Author Response · Authors · 2023-11-19
> **Official response to reviewer dokf(Part 3/4)**
>
> > w3 It is not apparent that why technical theoretical results are special for graphs. Many definitions and discussions seem to relate to general machine-learning problems. Can they be applied to data beyond graphs? If so, why the paper restrict the scope to graphs?
>
> While the underlying ideas considered in this paper are general and may be applicable to non-graphical tasks, the application of the specific techniques and arguments to more general domains is non-trivial and is an interesting avenue for future research. Particularly, the information-theoretic quantification of explainability of tasks and classifiers in Definitions 2 and 3, which is the foundation for our derivations in Theorem 1 and Propositions 1-3, does not naturally extend to non-graphical inputs. The reason is that the notion of explainability considered in this work is specific to sub-graph explanations. Recall that a graph $g_{exp}$ is a subgraph of $g$ if its vertices can be "aligned" with a subset of vertices in $g$ such that all edges between the two sets of vertices match. The focus on subgraph explainability is key in the definition of mutual information $I(Y;\overline{G}|{1}_{\Psi(\overline{G}})$ on Page 3 of the paper, which is used to define explainability in Definitions 2 and 3.
>
> To elaborate and provide additional explanation, let us consider an image classification task, where the objective is to detect the presence or absence of an object, e.g., a ball, in the input image. Consider a generic explanation method used in such image classification scenarios, such as SHAP (SHapley Additive exPlanations) and LIME (Local Interpretable Model-agnostic Explanations) methods. Both methods can be used to generate a "heat-map" signifying the influence of each of the (super-)pixels in generating the classification output. The heat-map can be used to choose the most important "k" (super-)pixels in the image, similar to how the most important edges of a graph are chosen by graph explanation methods considered in this work. The main question which needs to be addressed in order to make our derivations applicable to such a scenario is how to define a "sub-image" in an analogous way to the definition of a "subgraph". In that case, we can provide a more general version of Definitions 2 and 3 and rederive our results for other input domains.
>
> One method is to define two images sharing an explanation "sub-image" as two images that contain the explanation pixels at the exact same relative coordinates (both in the horizontal "x" and vertical "y" axis directions) as the explanation $g_{\exp}$, and then define $I(Y;\overline{G}|{1}\_{\Psi(\overline{G})})$ with respect to this definition. However, in this case, the explanation is fragile with respect to rotations and scaling. That is, if the image is rotated or scaled, the explanation would be considered a different explanation by this definition and would not affect the computation of $I(Y;\overline{G}|{1}\_{\Psi(\overline{G})}=g\_{exp})$. A direct result of this is that Condition 1 in Equation (1) would not hold. Which in turn implies Theorem 1 may not hold. As a result, in this scenario, many of the results which form the foundation of the paper should be re-derived with these new considerations. Note that Theorem 1 and Propositions 1-3 lead us to define our new fidelity measures, and provide theoretical guarantees for their robustness with respect to OOD issues in specific scenarios.
>
> It should be noted that, one can introduce an algorithm, similar to Algorithms 1 and 2, for image explanations, which operate as follows: i) Take an input image, ii) Apply a standard explanation mechanism such as LIME or SHAP, iii) take the highest rated $k$ (super-)pixels as explanation sub-image, iv) To compute $Fid_-$, keep the explanation image along with a sampled set of pixels from the original image, and use padding to fill-out the rest of the missing pixels, and to compute $Fid_+$, keep the non-explanation part of the image along with a sampled set of (super-)pixels from the explanation sub-image, and use padding to fill-out the rest of the missing pixels. The empirical evaluation of the performance of this algorithm is an interesting direction for future research. However, theoretical performance guarantees would require a non-trivial extension of the framework introduced here as described in the above paragraphs.
>
> We thank the reviewer for directing us toward studying the extensions of these ideas to non-graphical domains, and we plan to consider this problem in future works, in particular, to answer the question of how to define analogous structures to subgraphs for non-graphical explanations.

---

> ### Author Response · Authors · 2023-11-19
> **Official response to reviewer dokf(Part 4/4)**
>
> > Q1: Can the proposed fidelity measures be applied to data beyond graphs? If so, why the paper restrict the scope to graphs?
>
> As discussed in detail in response to Comment 3 in the weaknesses section, the arguments provided in this work do not extend naturally to non-graphical domains. Particularly, the notion of subgraphs, which is fundamental to our definitions and derivations needs to be developed for non-graphical domains. We will explore this in future works, and separately study explanations in non-graphical domains, as the reviewer has suggested.
>
> Should you have any further questions or require additional information, please feel free to contact us.

---

> ### Author Response · Authors · 2023-11-21
> **A kind reminder**
>
> Dear reviewer dokf,
>
> we submitted the reply a few days ago. Now the deadline for public comment is approaching. We are keen to ensure that our revisions and responses align with your expectations and address all your concerns effectively. Please feel free to let us know if you have other questions.

---

> ### Author Response · Authors · 2023-11-22
> **Final Discussion Time**
>
> Dear Reviewer dokf,
>
> The rebuttal phase ends today and I haven't received feedback from you.  We believe that we have addressed your previous concerns about the paper's organization and motivation. We have also added two experiments.  We would really appreciate that if you could check our response and updated paper.
>
> Looking forward to hearing back from you,
>
> Best Regards,
> Authors

---

### Author Response · Authors · 2023-11-22
**Waiting for Reviewers' Feedback**

Thank all reviewers for their comprehensive evaluations and insightful opinions on our paper.  We appreciate that our contributions have received positive recognition.  We have submitted an updated version of the manuscript in PDF, incorporating the suggestions provided by the reviewers.   In summary:

- We added a descriptive title to each of the Proposition and Theorem statements in the paper, including Theorem 1, Proposition 1, Proposition 2, and Proposition 3.
- We separated Section 3 into several subsections, and modified the subsection titles for Sections 2 and 3 to clearly reflect the contents of each subsection.
- We added more discussion for the motivation of Fidelity measures in the introduction (para 2 and 3).
- We added more discussion of differences and relationships to the existing explainable GNN methods in related work
- We added two more experiments on distribution analysis and case studies.

We believe that we have addressed the reviewer dokf's concerns about the paper's organization and motivation, and the reviewer gPjw's concern about the well-behaveness of the proposed method.

 We provided our response a few days ago.   As the author-reviewer discussion period is closing soon, we would appreciate it if reviewers could check our responses and updates. And we are happy to discuss and provide further evidence for any outstanding questions.

---

### Meta-Review · Area_Chair_gstu · 2023-12-05

**Metareview:**

This study examines existing fidelity metrics for assessing Graph Neural Network (GNN) explanation methods. The authors identify shortcomings in current metrics, attributing them to the out-of-distribution nature of explanation subgraphs. To address this, they propose modified fidelity metrics, transforming explanation subgraphs to approximate the underlying data distribution. The aim is to obtain more accurate predictions from trained GNN models and establish well-behaved proxy fidelity metrics. The paper emphasizes a critical problem in evaluating GNN explainability, addressing distribution shifts in subgraphs. The proposed information-theoretic definition and straightforward evaluation methods, applied to four datasets and two tasks, demonstrate improved consistency with ground truth explanations.

Overall, the proposed method is interesting and some reviewers are excited about the results. So, I also vote for acceptance.

**Justification For Why Not Higher Score:**

This is a borderline paper and it is good to present in poster.

**Justification For Why Not Lower Score:**

N/A

---

### Decision · Program_Chairs · 2024-01-16

Accept (poster)